# SCALABLE GANS WITH TRANSFORMERS

## ABSTRACT

Scalability has driven recent advances in generative modeling, yet its principles remain underexplored for adversarial learning. We investigate the scalability of Generative Adversarial Networks (GANs) through two design choices that have proven to be effective in other types of generative models: training in a compact Variational Autoencoder latent space and adopting purely transformer-based generators and discriminators. Training in latent space enables efficient computation while preserving perceptual fidelity, and this efficiency pairs naturally with plain transformers, whose performance scales with computational budget. Building on these choices, we analyze failure modes that emerge when naively scaling GANs. Specifically, we find issues as underutilization of early layers in the generator and optimization instability as the network scales. Accordingly, we provide simple and scale-friendly solutions as lightweight intermediate supervision and width-aware learning-rate adjustment. Our experiments show that the proposed Generative Adversarial Transformers (GAT), a purely transformer-based and latent-space GANs, can be easily trained reliably across a wide range of capacities (S through XL). Moreover, GAT-XL/2 achieves state-of-the-art single-step, class-conditional generation performance (FID of 2.18) on ImageNet-256 in just 60 epochs, 4× fewer epochs than strong baselines.

## 1 INTRODUCTION

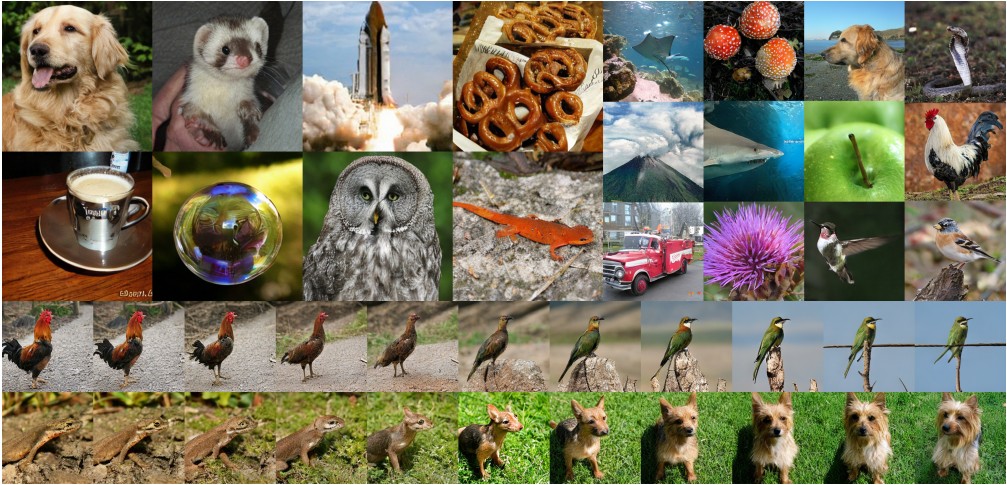

Figure 1: **Curated examples of GAT-XL/2 on ImageNet-256.** GAT-XL/2 exhibits strong generation capability (FID 2.18) within 60 epochs, 4× fewer than 1-NFE baselines (FID 3.43), while keeping the characteristics of GANs such as latent interpolation (bottom two rows).

Recent breakthroughs in generative modeling have become a central driver of progress across core areas of computer vision. These developments have accelerated in recent years, enabling capabilities that were previously out of reach: state-of-the-art systems now support text-to-image (Rombach et al., 2022; Podell et al., 2023; Esser et al., 2024b; Han et al., 2024) and text-to-video synthesis (Yang et al., 2024; Chen et al., 2024; Bar-Tal et al., 2024), demonstrate practical applica-

tions (Brooks et al., 2024; Google DeepMind, 2025c;b), and further enable the creation of 3D content (Zhao et al., 2025) and large-scale world simulation models (Google DeepMind, 2025a).

At the core of this advance is scalability: enlarging model capacity and data coverage reliably improves performance, often near-monotonically. When pushed to sufficiently large regimes, these trends yield marked gains in fidelity, coverage, and controllability. Crucially, these benefits depend on scale-friendly choices, including architectures that maintain stable signal flow, training recipes that remain well-behaved as width, depth, and batch size grow, and computational efficiency. Such scaling behavior has already been demonstrated in certain types of generative models such as autoregressive and diffusion families (Tian et al., 2024; Peebles & Xie, 2023; Liang et al., 2024).

By contrast, the scalability of Generative Adversarial Networks (GANs) has not been discussed yet, despite its attractive single-step sampling efficiency and interesting property of semantic latent space. While there have been attempts to train GANs at large scale (Kang et al., 2023; Zhu et al., 2025; Sauer et al., 2023), these efforts typically focus on a single high-capacity model with extensive, task-specific tuning, and thus do not constitute evidence of genuine scalability.

In this work, we revisit GANs in the aspect of scalability. We focus on two ingredients that have proven central to the success of scalable generative models. First, these models are typically trained in a low-dimensional latent space; the spatial latent grid produced by a pretrained, frozen VAE (Rombach et al., 2022) as an image tokenizer/de-tokenizer, enabling a dramatic reduction of the computational burden of both learning and inference while preserving high perceptual fidelity. Second, they employ transformer architectures, which are known for their scalability against width, depth, data, and compute.

Inspired by these two crucial factors, we combine these two elements to build a novel, scalable GAN framework: we construct a pure transformer-based GAN that operates in a compact latent space and study its behavior across substantial capacity ranges. We aim to assess the scalability of this design and to clarify the architectural and optimization choices. Accordingly, we pinpoint the hurdles that hinder adversarial training at scale. In detail, we identify the two key problems: (1) the early layers of the generator become inactive, leading to marginal contribution in image synthesis and (2) naïvely increasing depth and width with identical configuration leads to failures in convergence.

To address the first issue, we propose Multi-level Noise-perturbed image Guidance (MNG), which provides supervision at multiple intermediate layers of the generator. Specifically, we leverage a noise hierarchy: the synthesized images from earlier stages are trained to resemble the real data perturbed by a stronger image-level Gaussian, and the noise level monotonically decreases with depth. They serve as direct supervision for the generator's intermediate layers, restoring early-layer influence and improving layer-wise utilization throughout the network.

For the second issue, we focus on the fact that both the static initialization and optimization scheme amplify output magnitudes as the model grows deeper and wider. Specifically, as model size increases, the entire network tends to exhibit more rapid changes in its outputs per optimization step. This phenomenon implies that the training speed changes proportionally to the model scale, potentially causing instability in GAN training dynamics. Thus, we devise a simple scaling rule for adjusting the hyperparameters, especially the learning rate, to preserve the constant magnitude of changes in network output regardless of scale.

We experimentally validate that our framework, Generative Adversarial Transformers (GAT), is successfully trained on various scales of model (GAT-S to GAT-XL) and achieves FID of 2.18, which is the state-of-the-art performance in a one-step generation task on the class-conditional generation in ImageNet-256 dataset only within 60 epochs of training, while keeping the advantages of GAN, such as a single inference step or latent space manipulation (Fig. 1, more examples are available in Appendix).

## 2 PROPOSED METHOD

### 2.1 PRELIMINARIES

**Generative Adversarial Networks** Generative Adversarial Networks (GAN) (Goodfellow et al., 2014) is an adversarial learning framework between two networks, the generator $G(z, c)$ and dis-

criminator $D(I, c)$. Specifically, for a given randomly sampled latent code $z \in \mathbb{R}^{d_z} \sim p_z$ and condition $c$, the generator $G(z, c)$ synthesizes a fake image $\hat{x} \in \mathbb{R}^{H \times W \times 3}$ and the discriminator learns to distinguish the real image $x \in \mathbb{R}^{H \times W \times 3}$ and the fake image $\hat{x}$, while the generator learns to deceive the discriminator.

GAN has several interesting properties compared to other types of generative models, diffusion and AR models. For example, it offers extremely low dimensional latent space (e.g. $d_z = 64$) and semantic latent space which is suitable for image manipulation. Moreover, its generation process requires only a single inference step, making inference highly efficient. Despite these advantages, GAN has not been explored in terms of scalability, which is one of the main cause of the success of other generative model. In this paper, we study how to scale GAN using on the transformer architecture that is already verified its scalability across various tasks.

## 2.2 GENERATIVE ADVERSARIAL TRANSFORMERS

We introduce Generative Adversarial Transformers (GAT), a transformer-based GAN framework at the latent space of VAE, for the first time. Our primary goal is to preserve the design of transformer as much as possible to keep its scalability. Basically, we build GAT on the latent space of VAE (Rombach et al., 2022), following the recent advances in generative models (Rombach et al., 2022; Peebles & Xie, 2023; Tian et al., 2024). This allows us to efficiently increase the model size by reducing the computation costs of the generative model largely. For simplicity, we use the terms "VAE latent" and "image" interchangeably. In the following paragraphs, we describe our design of generator and discriminator architectures.

**Generator architecture**    Our generator adopts a standard Vision Transformer (ViT) architecture, consisting primarily of a stack of transformer blocks. Since the generator does not take input images, we remove the patchify layer and instead introduce an unpatchify layer (i.e., the RGB layer in Fig. 2) to synthesize images. Specifically, the unpatchify layer acts as a linear decoder, comprising normalization, linear projection, and reshaping operations. The output dimension of this linear decoder scales with the patch size $p$, increasing proportionally to $p^2$.

The transformer block (GAT block) follows the standard ViT design, but incorporates additional conditioning via the latent code $z$ and class condition $c$. Specifically, we employ a mapping network, a simple MLP, that generates a style vector $w$ from $z$ and $c$. This style $w$ is then used to modulate features through adaptive normalization and Layerscale (Touvron et al., 2021), drawing inspiration from StyleGAN (Karras et al., 2019) and DiT (Peebles & Xie, 2023). Note that Layerscale denotes a learnable, near-zero–initialized scale to each residual branch of attention and MLP block. Concretely, we produce scaling parameters $\gamma$ and $\alpha$ from $w$, which control the de-normalization and Layerscale (Touvron et al., 2021), respectively. Since we adopt RMSNorm, the shift parameter is omitted. To enhance stability during early training, both $\gamma$ and $\alpha$ are initialized to small values. Detailed explanations are provided in the Appendix. Throughout this paper, we refer to GAT as a "pure Transformer" generator in the sense that its backbone strictly follows a ViT/DiT-style architecture without convolutional components; the StyleGAN-inspired part is limited to a lightweight per-channel modulation on normalized features, similar to conditional LayerNorm/FiLM (Perez et al., 2018).

**Discriminator architecture**    The discriminator also adopts a Vision Transformer (ViT) backbone, with Layerscale applied to the output of each transformer block. As in the generator, the Layerscale parameters are initialized to small values to ensure stability during the early stages of training. To perform real/fake classification, a dedicated [cls] token is appended to the sequence of visual tokens before the first transformer block. This [cls] token is processed jointly with the other tokens and subsequently passed through a linear projection head to produce the discriminator logit.

## 2.3 ACTIVATING EARLY GENERATOR LAYERS VIA MULTI-LEVEL NOISE-PERTURBED IMAGE GUIDANCE

With the recent advances in GANs objectives (Huang et al., 2024), we observe that plain ViT-based generators and discriminators at the base scale can be trained successfully in the VAE latent space. However, analysis reveals that the early layers of the generator remain largely inactive. This means

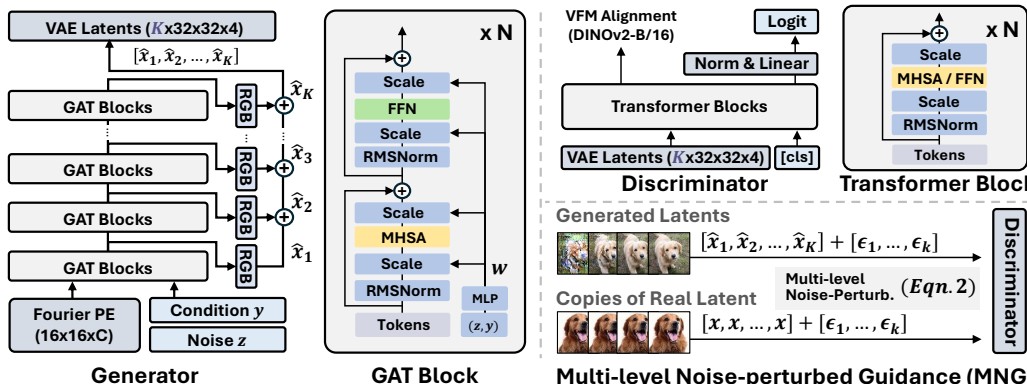

Figure 2: Generative Adversarial Transformers (GAT) architecture. Both the generator and discriminator are built from transformer blocks, augmented with modulation in $G$ and Layerscale in $D$. Our generator synthesizes auxiliary outputs from intermediate layers, which are paired with multiple noise levels and forwarded into the discriminator. Through supervision on intermediate outputs, this Multi-level Noise-perturbed Guidance (MNG) encourages all layers to contribute to images and consequently leverages the model capacity more effectively.

that their computations only marginally contribute to the final output, indicating the generator inefficiently utilizes its model capacity (Fig. 4). To address this inactivity of early layers, we draw inspiration from MSG-GAN (Karnewar & Wang, 2020), which introduces supervision on intermediate generator outputs (i.e., multi-scale supervision). We leverage its multi-level supervision with the explicit objective of increasing layer-wise contribution, particularly activating the early stages.

To this end, we propose the Multi-level Noise-perturbed image Guidance (MNG) strategy for training GANs. Firstly, we divide the generator into multiple $K$ stages and enforce auxiliary outputs at each stage. Each intermediate output is connected to the final synthesis path through residual connections, ensuring that information from early blocks is not discarded but accumulated across depth. Throughout this process, for the intermediate output $\hat{x}_k$ at $k^{\text{th}}$ stage, the output of the generator is defined as follows:

$$G(z, c) = [\hat{x}_1, \hat{x}_2, ..., \hat{x}_k]. \tag{1}$$

Then, we perturb each intermediate output $x_k$ by a Gaussian noise with a predefined noise strength. In detail, the pre-defined strengths build a hierarchy by assigning stronger noise perturbation to earlier stages and weaker corruption to later ones. After perturbation, all perturbed images are forwarded to the discriminator, guiding each generator stage to learn only the level of coarse structure that survives under its pre-defined noise. This process is defined as follows:

$$\mathcal{E}(\hat{x}_k; \alpha_k) = \alpha_k \hat{x}_k + \sqrt{1 - \alpha_k^2}\, \epsilon, \quad \alpha_1 < \alpha_2 < \cdots < \alpha_K, \quad \alpha_K = 1, \quad \epsilon \sim \mathcal{N}(0, I) \tag{2}$$

$$\ell = D(\mathcal{E}([\hat{x}_1, ..., \hat{x}_k]), c) = D([\mathcal{E}(\hat{x}_1), ..., \mathcal{E}(\hat{x}_k)], c), \tag{3}$$

where $\ell$ is the logit and $\hat{x}_k$ is the noised-perturbed counterpart of $x_k$ and $\alpha_k$ controls the degree of perturbation for noise-level $k$, increasing exponentially with depth. For simplicity, we omit the noise strength $a_k$ for the noise perturbing operator $\mathcal{E}$. Thus, earlier layers are supervised to match heavily noised images ($\hat{x}_1$), while later layers are aligned with clean targets ($\hat{x}_K$), forming a coarse-to-fine trajectory. For real data $x$, we use identical images for every level $k$.

This strategy encourages the early layers to capture global structure under strong noise corruption, while later layers progressively refine fine-grained details as the noise diminishes. By incorporating this multi-level noise supervision, applied through intermediate outputs of generator and discriminator-side perturbations, we ensure that all layers contribute actively to the synthesis process, mitigating the problem of inactive early layers. Our method introduces the coarse-to-fine generation process into pure transformer architectures without introducing explicit resolution hierarchies (i.e., multi-scale images). Importantly, this mechanism incurs only negligible computational overhead while improving network utilization, especially in early layers.

## 2.4 SCALING RULE FOR STABILIZING THE TRAINING OF GAN

Recent diffusion models such as DiT (Peebles & Xie, 2023) demonstrate scalability while adopting identical hyperparameters regardless of model size. In contrast, we find that simply increasing the model size under an identical configuration often leads to training divergence in GANs. This is problematic as the manual tuning of hyperparameters for every scale would severely undermine scalability. To address this, we propose a simple and principled scaling rule.

The key idea of the guiding principle is to maintain a consistent update magnitude across different model widths. In practice, when each layer input is normalized to unit variance (as ensured by normalization layers), the expected squared norm of the input grows linearly with the number of channels. Consequently, the update rate of the model becomes proportional to both the learning rate and the channel dimension. Since GAN training is known to be highly unstable and particularly sensitive to the choice of learning rate, preserving a constant update magnitude is crucial for preventing divergence and ensuring stable adversarial training dynamics. Therefore, when scaling up the model size, the learning rate should decrease inversely with the number of channels so that the overall update scale remains stable.

Formally, let $\eta_{base}$ denote the learning rate for the *base* model with channel size $C_{base}$, where the *base* model is the model that we tune the hyperparameters. For a model with channel size $C_{\text{model}}$, we define the learning rate adapted for this model $\eta_{\text{adapt}}$ as follows:

$$\eta_{\text{adapt}} = \eta_{base} \cdot \frac{C_{base}}{C_{\text{model}}}. \tag{4}$$

Our rule is conceptually related to the equalized learning rate (Karras et al., 2017) used in conventional GANs, which normalizes parameter updates to be invariant to the channel size. In architectures such as transformer-based generators and discriminators, where channel dimensions are approximately constant across layers, our global scaling rule yields a similar stabilizing effect while remaining easy to implement, without any changes in model implementation.

## 2.5 TRAINING OBJECTIVES

For adversarial learning, we deploy relativistic pairing loss (Jolicoeur-Martineau, 2018) with the approximated version of two-sided gradient penalty (Lin et al., 2025), following R3GAN (Huang et al., 2024). Specifically, this objective is denoted as follows:

$$
\begin{align}
\mathcal{L}_G^{\text{adv}} &= f(D(\mathcal{E}(G(z,c)),c) - D(\mathcal{E}(x),c)), \tag{5}\\
\mathcal{L}_D^{\text{adv}} &= f(D(\mathcal{E}(x),c) - D(\mathcal{E}(G(z,c)),c)), \tag{6}\\
\mathcal{L}_{\text{aR1}} &= \tfrac{1}{\sigma^2}||D(\mathcal{E}(x),c) - D(\mathcal{E}(x+\epsilon'),c)||^2, \tag{7}\\
\mathcal{L}_{\text{aR2}} &= \tfrac{1}{\sigma^2}||D(\mathcal{E}(G(z,c)),c) - D(\mathcal{E}(G(z,c)+\epsilon'),c)||^2, \tag{8}
\end{align}
$$

where $f(\cdot)$ is a softplus function and $\epsilon' \sim \mathcal{N}(0, \sigma I)$ is a gaussian noise with a std $\sigma$.

In addition, inspired by the rationale of feature-aided GANs (Sauer et al., 2021; Kumari et al., 2022) and recent diffusion work on representation alignment (Yu et al., 2024), we encourage the discriminator to learn semantically rich Vision Foundation Models (VFM) features. Different from prior work (Yu et al., 2024), we do not use the generator for alignment, as $G$ takes noise as input and it is difficult to obtain VFM features directly from the generated (fake) data. Let $\phi(\cdot)$ be a frozen vision foundation model (e.g., DINOv2 (Oquab et al., 2023)), and let $H_D(x) = \{h_{\text{cls}}, h_1, \ldots, h_N\}$ denote the discriminator's $[\text{cls}]$ token and $N$ patch tokens at the last layer. We obtain teacher tokens $\hat{H}_\phi(x) = \{\hat{h}_{\text{cls}}, \hat{h}_1, \ldots, \hat{h}_N\}$ by forwarding the same image through $\phi$. Then, this alignment objective is defined as follows:

$$\mathcal{L}_{\text{REPA}} = \frac{1}{N+1} \sum_{i \in \{\text{cls}, 1:N\}} (\text{sim}(P(h_i), \hat{h}_i))). \tag{9}$$

Note that, this alignment objective is only applied with a real data, and $P$ denotes a small learnable MLP to align token dimensions, and sim is a similarity measure such as cosine similarity.

Table 1: **Class-conditional generation on ImageNet-256×256 (FID-50K)**. (Left) 1 or 2 Number of Function Evaluation (NFE) generative models. (Right) Other generative models including autoregressive models and multi-step diffusion/flow models. Diffusion/flow entries are reported under CFG, when applicable. Across both tables, '×2' denotes that CFG yields 2 NFEs for each sampling step. [†]: Leveraging ImageNet-pretrained discriminators, lowering FID more than the actual image quality (Kynkäänniemi et al., 2022).

| Method | Params | NFE | Epoch | FID | Method | Params | NFE | FID |
|---|---|---|---|---|---|---|---|---|
| ***2-NFE diffusion/flow from scratch*** | | | | | **autoregressive/masking** | | | |
| iCT-XL/2 | 675M | 2 | - | 20.30 | AR w/ VQGAN | 227M | 1024 | 26.52 |
| iMM-XL/2 | 675M | 1×2 | 3840 | 7.77 | MaskGIT | 227M | 8 | 6.18 |
| MeanFlow-XL/2 | 676M | 2 | 240 | **2.93** | VAR-$d$30 | 2B | 10×2 | 1.92 |
| ***1-NFE diffusion/flow from scratch*** | | | | | MAR-H | 943M | 256×2 | 1.55 |
| iCT-XL/2 | 675M | 1 | - | 34.24 | | | | |
| Shortcut-XL/2 | 675M | 1 | 250 | 10.60 | ***diffusion/flow*** | | | |
| MeanFlow-XL/2 | 676M | 1 | 240 | 3.43 | ADM | 554M | 250×2 | 10.94 |
| ***1-NFE GANs from scratch*** | | | | | LDM-4-G | 400M | 250×2 | 3.60 |
| StyleGAN-XL[†] | 166M | 1 | - | 2.30 | SimDiff | 2B | 512×2 | 2.77 |
| BigGAN | 112M | 1 | - | 6.95 | DiT-XL/2 | 675M | 250×2 | 2.27 |
| GigaGAN | 569M | 1 | 480 | 3.45 | SiT-XL/2 | 675M | 250×2 | 2.06 |
| GAT-XL/2 | 602M | 1 | **40** | **3.02** | SiT-XL/2+REPA | 675M | 250×2 | **1.42** |
| GAT-XL/2 | 602M | 1 | **60** | **2.18** | | | | |

In short, the full discriminator and generator objectives are

$$\mathcal{L}_D = \mathcal{L}_D^{\text{adv}} + \lambda_{\text{aGP}}\mathcal{L}_{\text{aR1}} + \lambda_{\text{aGP}}\mathcal{L}_{\text{aR2}} + \lambda_{\text{REPA}}\mathcal{L}_{\text{REPA}}, \qquad \mathcal{L}_G = \mathcal{L}_G^{\text{adv}}, \qquad (10)$$

where $\lambda_{\text{aGP}}$ and $\lambda_{\text{REPA}}$ are the strength of gradient penalty and alignment objectives, respectively. For other details, we further elaborate them in Appendix.

## 3 EXPERIMENTS

**Experimental settings.** We conduct all experiments with class-conditional generation on ImageNet (Deng et al., 2009) at a resolution of 256×256. For the evaluation metric, we mainly use Frechèt Inception Distance (FID) (Heusel et al., 2017) on 5K and 50K images. In line with standard practice, we employ the pre-trained Stable Diffusion variational autoencoder (SD-VAE) (Rombach et al., 2022) as a tokenizer for mapping between pixel and latent spaces. Accordingly, we train all models at a VAE latent spatial resolution of 32×32, as SD-VAE's downsample ratio is 8. Also, we evaluate four model capacities, Small (S), Base (B), Large (L), and XLarge (XL), following previous work (Peebles & Xie, 2023). We mainly perform experiments with patch size p=2. Each model is named by its model and patch size; for example, GAT-S/2 for small model with a patch size of 2.

We use identical hyperparameters for every scale of models except the learning rate, which we adaptively modify as elaborated in Sec. 2.4 . For class conditioning of discriminator, we use the projection discriminator (Miyato & Koyama, 2018). Basically, we instantiate the generator and discriminator with identically sized transformer backbones for each capacity. Every model is trained at a training budget of 50K iterations with a 512 batch size, same as 20 epochs in ImageNet dataset, and evaluated without the truncation trick or guidance (Zhang et al., 2024), unless specified.

### 3.1 COMPARISON WITH PRIOR ARTS

We compare the proposed method with various types of generative models, including one or two-step and multi-step GAN/diffusion/flow models. As reported in Tab. 1, our GAT-XL/2 achieves the state-of-the-art FID-50K on ImageNet-256, significantly enhancing the FID on 1-step generation (3.43 to 2.18). Notably, it reaches this performance with only 60 epochs, substantially fewer training epochs than prior methods. This experimental result implies strong data efficiency of the proposed method and suggests further gains can be achieved with longer training. More importantly, it shows that GANs possess generative capabilities that are not significantly inferior to those of other generative models.

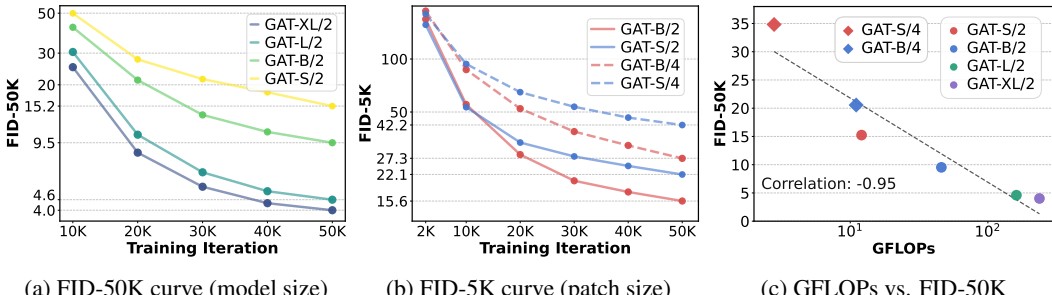

(a) FID-50K curve (model size)  (b) FID-5K curve (patch size)  (c) GFLOPs vs. FID-50K

Figure 3: Scalability of GAT. (a) Training curve of FID-50K across the various model sizes shows that the performance is monotonically increasing as the model size is scaled up. (b) Training curve of FID-5K across the various patch sizes. With an identical number of parameters, we observe that the higher computational power of models enhances the generation capability. (c) We observe strong negative corrlatin between FID-50K and GFLOPs, proving that the models with higher compute systematically yield better FID.

## 3.2 TRAINING GAT ON VARIOUS SCALES

**Model size.** We trained GAT across various model capacities, then measured the FID-50K for every 10K iterations. As shown in Fig. 3a, we observe that larger models consistently achieve lower FID, and this advantage mostly persists throughout training rather than appearing only at convergence. This scaling behavior shows that the training GAN can be easily scaled up, similar to other types of generative models, with minimal modification in hyperparameters.

**Patch size.** We further assess the robustness of the proposed method against tokenization granularity by performing experiments with a larger patch size of p=4 for the Small and Base configurations. As shown in Fig. 3b, the models are successfully trained and attain acceptable FID across patch sizes, indicating that the proposed method can be easily extended across various patch sizes.

**GFLOPs.** Model complexity is commonly measured by GFLOPs. Therefore, we also plot FID-50k against the transformer's computational cost measured in GFLOPs, and compute the correlation between the model's performance and its GFLOPs. As shown in Fig. 3c, we observe a strong negative correlation (-0.95): models with higher compute systematically yield better (lower) FID. These results indicate that scaling improves performance and that the proposed GAT is scalable and effectively utilizes the scalable characteristics of transformer architectures. Note that, GFLOPs are computed for a single forward pass of the generator.

## 3.3 ABLATION STUDY

**Multi-level Noise-perturbed image Guidance (MNG) (Sec 2.3).** As discussed earlier, we first demonstrate that a vanilla GAT without MNG displays inactive features in early layers. Accordingly, we perform a block-level analysis while ablating MNG. To this end, we visualize intermediate features for each transformer block using PCA. As shown in Fig. 4a, early-layer features are highly redundant without MNG, indicating that most early layers remain inactive. In contrast, our method yields well-distributed feature activations throughout the entire network.

As shown in Fig. 4b, to measure per-block influence, we ablate each transformer block, re-synthesize the image, and compute the LPIPS (Zhang et al., 2018) distance to the unablated output; smaller LPIPS implies a lower perceptual contribution on the generated images. We compute these statistics on 10K images. Aligned with the above observation, the model without MNG exhibits weak early-layer contribution on the generated images, that is, most of the generative process is concentrated in the later blocks. By contrast, our model shows a progressively decreasing contribution from early to late layers, which is precisely consistent with MNG's objective of coarse-to-fine synthesis: intermediate layers receive sufficient guidance, responsibility is distributed across depth, and the network's capacity is utilized more uniformly. Note that the last layer tends to spike, likely because it is located directly before the final synthesis result.

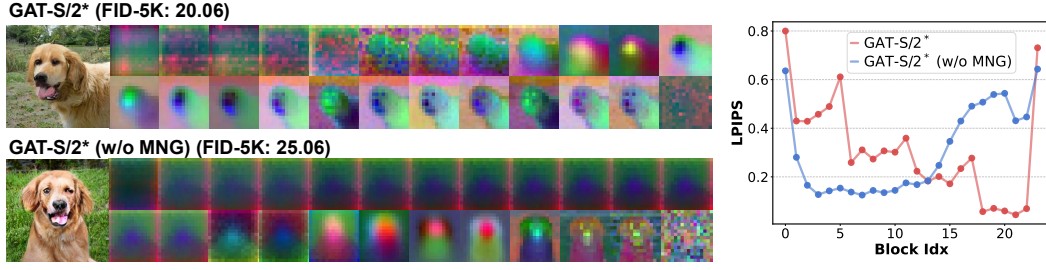

GAT-S/2* (FID-5K: 20.06)

GAT-S/2* (w/o MNG) (FID-5K: 25.06)

(a) Feature visualization by PCA top 3 components

(b) Effect of each block on LPIPS

Figure 4: Visualization of intermediate features of the generator and their effects on the generated images. (a) Both GAT models reveal the coarse-to-fine synthesis process, but without the Multi-level Noise-perturbed image Guidance (MNG), the generator's early layers become largely inactive, showing feature visualizations change only marginally, whereas our method activates these layers much earlier. (b) LPIPS distances while ablating Transformer blocks one by one. Without MNG, removing early blocks yields only minor changes in the output, despite those blocks producing coarse information, indicating computational inefficiency in the generator's early layers. GAT-S/2* doubles the number of blocks relative to GAT-S/2 for finer block-level analysis.

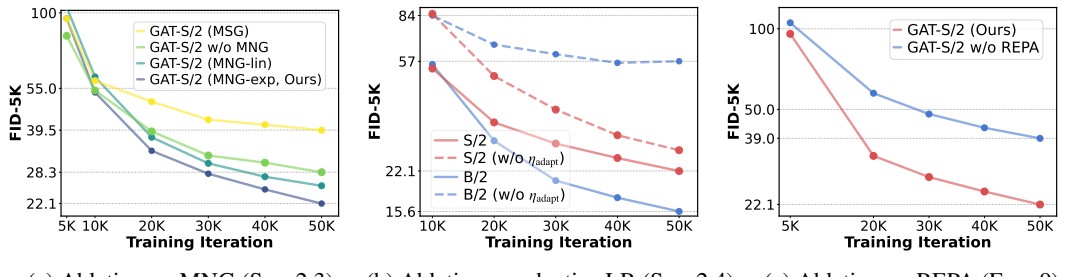

(a) Ablation on MNG (Sec. 2.3)  (b) Ablation on adaptive LR (Sec. 2.4)  (c) Ablation on REPA (Eqn. 9)

Figure 5: Ablation study. (a) Multi-level Noise-perturbed image Guidance (MNG) consistently enhances the performance throughout the entire training (vs. w/o MNG) and also surpasses the original MSG-GAN, which degrades images by resize operation (vs. MSG). (b) Effect of adaptive learning rate scaling. Each model converges stably with its own $\eta_{\text{adapt}}$, while transferring it with another model's $\eta$ leads to severe degradation. (c) The REPA objective substantially improves performance, indicating that advances from diffusion models can transfer effectively to GAT.

Furthermore, we evaluate MNG in a quantitative way. We plot the FID-5K training curves in Fig. 5a. We evaluate four variants: (i) MSG (replacing noising-based degradation with resize-based degradation, following MSG-GAN (Karnewar & Wang, 2020)), (ii) w/o MNG, (iii) MNG-lin (linear noise schedule), and (iv) MNG-exp (exponential noise schedule, our default setting). Across runs, our base setting, MNG-exp, consistently achieves the best (lowest) FID, outperforming both the no-MNG baseline and the linear schedule. Interestingly, MSG delivers the weakest performance. We hypothesize that, as reported in prior work (Lin et al., 2021; Kang et al., 2023), feeding the discriminator multi-scale outputs can overemphasize cross-scale consistency, which in turn suppresses generative quality. In contrast, our MNG perturbs a single degraded counterpart with stochastic noise at multiple levels, providing diversity without enforcing strict cross-scale alignment, and thereby avoiding the aforementioned failure mode.

**Adaptive learning rate (Sec. 2.4).** For each model, an appropriate learning rate is determined by the adaptive learning rate strategy (Fig. 5b), which ensures stable convergence. To assess the effectiveness of this strategy, we conduct a cross-check experiment by reusing configurations across scales (i.e., training GAT-S/2 with the $\eta_{\text{adapt}}$ of GAT-B/2; and vice versa). In this naive setting where we reuse the configuration of another model, performance degrades substantially: GAT-S/2 converges slowly due to an overly small learning rate, while GAT-B/2 diverges under an excessively large learning rate. These results indicate that our adaptive learning rate strategy reliably selects a proper learning rate across scales without any manual tuning, a key factor for scalability.

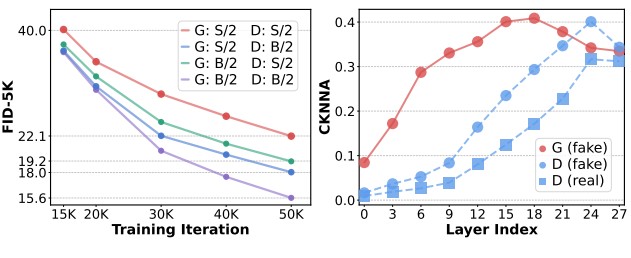

(a) Decoupled scaling of G and D    (b) CKNNA of G and D

Figure 7: Further analysis. (a) Scaling G and D separately shows both impact FID, while scaling D is more effective than G. (b) Feature alignment against DINOv2-g measured by CKNNA using real and fake data. We observe that the features obtained from fake data show higher alignment with VFM than real data.

**VFM alignment objective $\mathcal{L}_{\text{REPA}}$ (Eqn. 9).** We ablate the REPA objective, which aligns discriminator representations with those from a Vision Foundation Model (VFM), as in Fig. 5c. REPA significantly and consistently enhances the performance of the generator, although we impose a feature alignment objective only on the discriminator. Furthermore, this result implies that recent techniques developed for diffusion models using VFMs (Yao et al., 2025; Chen et al., 2025) can transfer effectively to our GAT framework.

### 3.4 FURTHER ANALYSIS

**Decoupled analysis of Generator and Discriminator scaling.** We analyze the relative contributions of $G$ and $D$ by scaling them individually. As shown in Fig. 6a, training remains stable and performance improves in both cases, but the gains from scaling the discriminator are notably larger. This suggests that, because the generator only learns through the discriminator's feedback, overall performance is effectively bottlenecked by how well the discriminator models the data distribution and shapes the real–fake decision boundary, so scaling up the discriminator, thereby providing sharper and more informative gradients, yields larger gains than merely increasing the generator's capacity. In addition, this observation aligns with our discussion below on the importance of representation learning in discriminators, highlighting its central role in adversarial learning.

**Representation Alignment of Generator and Discriminator.** Recent work on diffusion models (Yu et al., 2024) shows that generation quality tends to be proportional to the degree of feature alignment to Vision Foundation Models (VFMs). Motivated by this, we evaluate the feature-alignment metric CKNNA (Huh et al., 2024) of both the generator and discriminator against DINOv2-g on real and fake data (Fig. 6b). Our intuition is that generated samples tend to fall within the discriminator's well-established feature space, where representations are most reliable. In this space, the discriminator can provide strong and effective guidance, from which the generator consistently benefits, leading to higher-quality synthesis. Accordingly, as the generative performance of $G$ is tightly coupled with the representation learning ability of $D$, further strengthening discriminator representations may be a promising direction for future work.

## 4 RELATED WORKS

**Generative Adversarial Networks (GANs)** are trained through an adversarial game between a generator and a discriminator. The progress is mainly driven by architectural innovations and improved objectives. Architecturally, advances have largely come from convolutional models, especially the StyleGAN family (Karras et al., 2019; 2020), later extended to large-scale text-to-image generation (Kang et al., 2023; Sauer et al., 2023), though still limited to pixel space generation. Transformer-based approaches have also been explored (Jiang et al., 2021; Zhao et al., 2021; Lee et al., 2021), but their reliance on complex modification from plain transformer architectures and heavy hyperparameter tuning limits scalability. On the objective side, many adversarial losses (Goodfellow et al., 2014; Arjovsky et al., 2017; Lim & Ye, 2017; Mao et al., 2017) and regularization schemes (Mescheder et al., 2018; Gulrajani et al., 2017) have been proposed, with R3GAN (Huang et al., 2024) recently combining gradient penalties with a relativistic objective for greater stability. In parallel, a family of methods explicitly studies discontinuous discriminator functions and gradient normalization, such as GraN-GAN (Bhaskara et al., 2022) and gradient normalization (Wu et al., 2021), further improving stability and robustness. In this work, we establish a GAN framework in the latent space of a VAE, adopt a fully transformer-based design, and provide an empirical study of its scalability.

**Scalability of generative models** is a key factor in recent breakthroughs. Diffusion and flow models have demonstrated clear gains from transformer backbones (Peebles & Xie, 2023; Ma et al., 2024) and systematic scaling with data and compute (Liang et al., 2024), with latent-space tokenizers (Rombach et al., 2022; Yao et al., 2025; Chen et al., 2025), enabling efficient training and high-resolution synthesis (Esser et al., 2024b; Podell et al., 2023). Likewise, autoregressive models also have benefited from transformer scaling leading to substantial advances in generation quality in various domains, from class-conditional image generation to text-to-image synthesis (Chang et al., 2022; Tian et al., 2024; Han et al., 2024). In this work, we revisit the GANs framework through transformer-based latent architectures, which preserve single-step inference while inheriting the favorable scaling behavior of transformers.

## 5 CONCLUSION

We revisit GAN scalability by pairing VAE-latent training with plain transformer generators and discriminators. Addressing early-layer underuse and scale-coupled instability with lightweight intermediate supervision and width-aware learning-rate scaling yields GAT, which trains reliably from S to XL and reaches state-of-the-art one-step ImageNet-256 in 60 epochs (4× fewer than strong baselines). We hope our work will serve as a strong step forward in the potential of scaling GANs.

## 6 REPRODUCIBILITY STATEMENT

We provide the experimental settings and detailed hyperparameters in Sec. 3 and Appendix A.1. Also, we plan to release our code and pretrained model checkpoints for reproducibility.

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

# A APPENDIX

## A.1 IMPLEMENTATION DETAILS

We provide the configurations for all model sizes, including the parameter counts of the generator and discriminator in Fig. 2. Also, we report the detailed FID-50K score at 50K iterations in Tab. **??**, which is used for visualizing Fig. 3c.

**Generator** We design our models following common conventions from ViT (Dosovitskiy et al.) and StyleGAN (Karras et al., 2020). We use a latent code $z$ of dimension $d_z = 64$, and initialize the class embedding with the standard ViT token scale of 0.02. The mapping network is a shallow MLP whose width matches the transformer hidden dimension; it consists of two linear layers with a single nonlinearity, using SiLU in line with transformer practice.

Following StyleGAN, we train the mapping network with a learning rate that is $100\times$ smaller than the rest of the generator. The main GAT block is as described in the paper, and we additionally adopt techniques reported to improve transformer performance, Rotary Positional Embeddings (RoPE) (Su et al., 2024), SwiGLU-FFN (Shazeer, 2020), and qk-normalization. Finally, all scaling parameters produced from style codes are initialized to have a variance 0.1.

For the number of intermediate outputs $K$, we use $k = 4$ for every model size. These outputs are synthesized at uniform intervals across the generator's GAT blocks. For example, in GAT-XL/2 with 28 layers, we take an output every 7 layers.

**Discriminator** The discriminator largely follows a standard ViT, with the sole exception that each module output is gated by a Layerscale factor; all Layerscale vectors are initialized to 0.1. Similar to the generator, every transformer block uses RoPE, a SwiGLU feed-forward network, and qk-normalization. The projection layer, for the VFM-alignment objective, $P$ follows REPA (Yu et al., 2024) and is implemented as a 3-layer MLP with a hidden dimension of 2048. Also, we deploy DINOv2-B as a vision foundation model to align with.

During training, we apply differentiable augmentation (Zhao et al., 2020). To combine it with the noise-adding operations (approximated GP and multi-level noise-perturbation guidance), we proceed as follows: upon receiving an input image, we first add the perturbation used for the approximated GP, then apply the augmentation, and finally apply the multi-level noise perturbations. For the approximated GP, the same noise magnitude is used for all noise levels ($\sigma = 0.01$).

**Noise sampling and schedule for MNG** We design the image signal doubles at each successive output. Since the final output should be a clean image, for $k = 4$ we set

$$(\alpha_1, \alpha_2, \alpha_3, \alpha_4) = (0.125, 0.25, 0.5, 1.0).$$

In addition, we build the noise at each level cumulatively, starting from the noise added to the clean image and accumulating the newly sampled noise for constructing lower-level noise.

Given noise $\epsilon_k$ at level $k$, we obtain the noise $\epsilon_{k-1}$ at level $k - 1$ as follows:

$$\epsilon_{k-1} = r_k \, \epsilon_k + \sigma_k \, \eta_k, \qquad \eta_k \sim \mathcal{N}(0, I),$$

where the signal schedule is $\alpha_1 < \cdots < \alpha_K$ with $\alpha_K = 1$, and

$$r_k = \frac{\alpha_{k-1}}{\alpha_k}, \qquad \sigma_k = \sqrt{1 - r_k^2}.$$

This noise sampling preserves the variance of $\epsilon_k$ at every level while keeping the noise already sampled at the higher levels.

**Other hyperparameter** Basically, every hyperparameter is shared across any size of models, except the learning rate. We train with a gradient-penalty coefficient $\lambda_{\text{aGP}} = 1 \times 10^{-1}$ and VFM alignment objective coefficient $\lambda_{\text{REPA}} = 1$. The optimizer is AdamW with $(\beta_1, \beta_2) = (0.0, 0.99)$ (following common GAN practice such as StyleGAN). We apply exponential moving average (EMA) to the generator with decay 0.999. Also, we use a batch size of 512, `bfloat16` precision, gradient checkpointing, and PyTorch Scaled Dot-Product Attention (SDPA) implementation.

Table 2: Model configuration and parameter counts (M = million).

| Model | Layers | Dim | Heads | G params | D params |
|---|---|---|---|---|---|
| GAT-S | 12 | 384 | 6 | 29.36M | 39.21M |
| GAT-B | 12 | 768 | 12 | 116.75M | 104.68M |
| GAT-L | 24 | 1024 | 16 | 408.75M | 323.04M |
| GAT-XL | 28 | 1152 | 16 | 602.25M | 467.68M |

Table 3: FID at 50K iter. across model sizes.

| Model | FID-50K |
|---|---|
| GAT-XL/2 | 4.021 |
| GAT-L/2 | 4.600 |
| GAT-B/2 | 9.534 |
| GAT-S/2 | 15.237 |

Table 4: Ablation on MNG (FID-5K).

| Model | FID-5K |
|---|---|
| GAT-S/2 ($lr$=4e-4) | 22.080 |
| GAT-B/2 ($lr$=2e-4) | 15.610 |
| GAT-S/2 ($lr$=2e-4, w/o $\eta_{adapt}$) | 26.410 |
| GAT-B/2 ($lr$=4e-4, w/o $\eta_{adapt}$) | 56.720 |

For learning rate, we use $4 \times 10^{-4}$ as the *base* learning rate for the GAT-S model. After applying our adaptive learning rate rule, the per-size learning rates are: (GAT-S, GAT-B, GAT-L, GAT-XL) = $(4 \times 10^{-4}, 2 \times 10^{-4}, 1.5 \times 10^{-4}, 1.33 \times 10^{-4})$.

**Compute resource** For our largest experiment, training GAT-XL/2 within 40 epochs in ImageNet-256 dataset requires about 12 days with $8 \times$NVIDIA RTX A6000 GPU.

## A.2 ADDITIONAL RELATED WORKS

We simply explain the baselines that we compare with in Tab. 1.

- **VQGAN** (Esser et al., 2021) introduce the GPT-like autoregressive model on the discretized visual tokens to build the generative model.

- **ADM** (Dhariwal & Nichol, 2021) proposes the U-Net-based diffusion architecture with a classifier guidance, firstly beating the GAN counterpart in image generation task.

- **MaskGIT** (Chang et al., 2022) proposes a parallelized decoding strategy to improve the inference speed of autoregressive models.

- **LDM** (Rombach et al., 2022) proposes to train diffusion model on the latent space of pre-trained VAE, enhancing the generation capability and inference speed.

- **SimDiff** (Hoogeboom et al., 2023) improves the standard denoising diffusion model to train directly in pixel space on high-resolution images.

- **DiT** (Peebles & Xie, 2023) proposes replacing the conventional U-Net backbone in diffusion models with plain (non-hierarchical) transformers with AdaLN-zero layer.

- **iCT** (Song & Dhariwal, 2023) introduces distillation-free consistency training recipe, which surpasses previous consistency distillation.

- **SiT** (Ma et al., 2024) conducts an in-depth study showing that transitioning from discrete diffusion to continuous flow matching makes DiT training more efficient.

- **VAR** (Tian et al., 2024) introduces visual autoregressive model that substitutes spatial autoregression with progression across scales.

- **MAR** (Li et al., 2024) proposes a framework for training autoregressive models on continuous tokens by introducing a shallow diffusion model to sample the next token.

- **Shortcut** (Frans et al., 2024) learns the shortcut between two apart timestep to predict the single-step denoising direction.

- **iMM** (Zhou et al., 2025) proposes the method to train few-step generators from scratch by using self-consistent interpolants and matching all moments along the data.

- **MeanFlow** (Geng et al., 2025) introduces one-step generative framework which predicts average velocity, the time integral of the instantaneous velocity.

### A.3 EXACT FID VALUES FOR FIG. 3C AND FIG. 5A

Fig. 3c and Fig. 5a present the GFlops vs. FID-50K comparison and the ablation results on MNG, respectively. To improve clarity, we provide the exact FID scores corresponding to these plots. Specifically, Tab. 3 lists the FID values used in Fig. 3c, and Tab. 4 reports the FID results for the MNG ablation in Fig. 5a.

### A.4 TRAINING CURVE OF GAT-XL/2 (FID-50K)

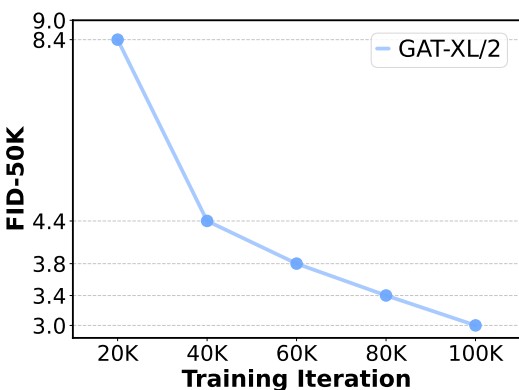

Figure 8: Training curve of GAT-XL/2 until 40 epochs.

We additionally report the FID-50K training curve for GAT-XL/2 up to 100K iterations (i.e., 40 epochs). The metric decreases monotonically, suggesting that further training would likely yield additional improvements.

### A.5 EXTENDED TRAINING ON IMAGENET-256 WITH GAT-XL/2

Table 5: Comparison between GAT-XL/2 and Diffusion Transformer (DiT) on ImageNet-256. Our GAT-XL/2 achieves a lower FID-50K score under substantially fewer training epochs.

| Model | Dataset | Epochs | FID-50K |
|---|---|---|---|
| GAT-XL/2 (ours) | ImageNet-256 | 60 | 2.18 |
| DiT-XL/2 (Peebles & Xie, 2023) | ImageNet-256 | 1400 | 2.27 |

To probe the performance ceiling of GAT-XL/2, we extended training beyond the reported 40 epochs. Specifically, we increased the gradient-penalty weight to $1 \times 10^1$ and trained for an additional 20 epochs, after which GAT-XL/2 achieved an **FID-50K of 2.18** on ImageNet-256. This result *not only* improves upon our original 40-epoch performance, but also *surpasses* the FID 2.27 reported for the Diffusion Transformer (DiT-XL/2) (Peebles & Xie, 2023), which requires 1400 training epochs on the same dataset, thereby achieving better quality with over an order-of-magnitude fewer epochs. Moreover, as a single-step generator, GAT-XL/2 also enjoys substantially faster inference: compared to the diffusion transformer, which typically requires around 250 denoising steps, our model attains comparable or better FID with $250\times$ fewer function evaluations (NFE) per sample.

### A.6 TRAINING TIME AND INFERENCE COST OF GAT

**Training time** Because GAT updates both the generator and discriminator at every iteration, a single training step is expensive than that of a purely generative diffusion model. In our measurements, processing the same number of images takes roughly $5\times$ longer per iteration compared to a DiT-style diffusion transformer. However, this does *not* imply a longer overall time-to-quality. For example, DiT-XL/2 requires about 1400 epochs on ImageNet-256 to reach FID 2.27, which is reported to take roughly 36 days on $8\times$A100 GPUs (Zheng et al., 2023). In contrast, our GAT-XL/2 attains a better FID-50K of 2.18 after only 60 epochs, corresponding to approximately 18 days on

a less powerful setup with $8\times$RTX A6000 GPUs. In addition to the lower wall-clock training time, this also means GAT observes fewer total passes over the data, indicating better data efficiency. Importantly, the modern one-step diffusion model, such as MeanFlow (Geng et al., 2025), also requires additional computation of the gradient, so it also requires additional time.

Furthermore, most of this training cost is dominated by the discriminator, which must repeatedly distinguish real from fake samples and evaluate the gradient-penalty term. We therefore expect that the pretrained feature extractor as a discriminator, in the spirit of projected GANs (Sauer et al., 2021), and then fine-tuning only the later layers for real/fake discrimination could substantially reduce this cost, as follows previous GANs (Sauer et al., 2022; 2023). Exploring such pretrained discriminator and its impact on training time and stability is an interesting direction for future work.

Table 6: Inference latency and memory footprint when generating 64 ImageNet-256 samples on a single Titan RTX GPU.

| Model | NFE | Time / image (s) | Peak memory (MB) |
|---|---|---|---|
| DiT-XL/2 | 250 | 15.2612 | 4525.85 |
| GAT-XL/2 (ours) | 1 | 0.0773 | 6028.11 |

**Inference cost.** GAT shares almost the same pure transformer backbone as DiT (Peebles & Xie, 2023), since we deliberately keep the architecture close to a plain transformer block. Consequently, the *per-step* inference cost and memory footprint for a single NFE are very similar between GAT and DiT at matched width and depth. The key difference comes from the number of function evaluations (NFE): while DiT typically requires around 250 denoising steps, GAT is a single-step generator (1 NFE). In practice, this translates into roughly two orders of magnitude speedup; in our measurements (Table. 6), GAT-XL/2 is about $200\times$ faster than DiT-XL/2 at comparable quality.

Furthermore, most of the end-to-end inference memory in both models is dominated by the transformer backbone. Since GAT performs truncation and guidance directly in the latent space before the backbone, it does not require additional passes through the heavy network, so the backbone-side memory usage remains essentially comparable to DiT. Any residual difference in peak VRAM mainly comes from lightweight auxiliary heads (e.g., MNG feature caching), rather than from the core architecture.

Concretely, we measure inference latency and memory on a single Titan RTX GPU when generating 64 ImageNet-256 samples, as shown in Table. 6. For DiT-XL/2, using the standard 250-step sampler, generating one image takes 15.2612 seconds with a peak memory usage of 4.5 GB. In contrast, GAT-XL/2 with 1-NFE generation requires only 0.0773 seconds per image, corresponding to roughly a $200\times$ speedup, with a peak memory usage of 6.0 GB. The slightly higher VRAM footprint for GAT is mainly due to caching intermediate features for the MNG guidance head; the transformer backbone itself has a comparable memory cost to DiT at similar width and depth.

### A.7 MOTIVATION FOR MULTI-LEVEL NOISE-PERTURBED GUIDANCE

**Motivation** The central motivation for MNG comes from how other generative models couple their inputs with the final generated image that takes the supervision. In diffusion and autoregressive models, the input and output at each step are tightly linked by the training objective: step-wise denoising in diffusion, or next-token prediction in autoregressive models. This step-wise supervision explicitly enforces a strong relationship between intermediate step inputs and clean targets throughout the depth of the network, so that intermediate representations must remain informative with respect to the final sample.

In contrast, a standard GAN generator typically receives a low-dimensional latent code that is mapped to a constant input, while the adversarial loss is applied only to the final image via the discriminator. As long as the last few blocks can synthesize images that fool the discriminator, there is little incentive for earlier blocks to maintain rich structure or to remain sensitive to the latent code. Optimization can therefore push most of the representational burden toward the blocks closest to the output, leaving early layers underutilized or close to constant mappings, as observed in our empirical analysis.

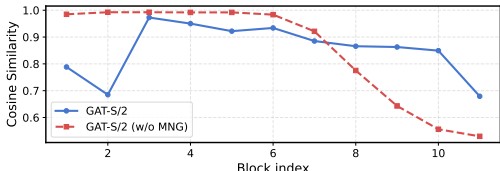 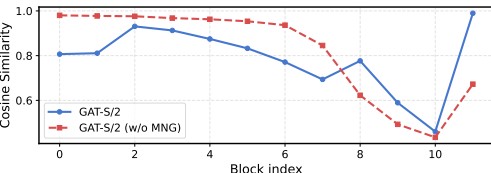

(a) Feature similarity between block input and output. (b) Feature similarity across different input noises $z$.

Figure 9: Quantitative analysis of early-layer inactivity with and without MNG. (a) Cosine similarity between the input and output features of each block, averaged over all spatial locations, tokens, and images. (b) Cosine similarity between features obtained from different input noises $z$ for the same class. Without MNG, both similarities are close to 1 in the early blocks, indicating that they induce only minor feature changes and are weakly influenced by the input noise. With MNG, the similarities in early blocks are reduced for both measures, showing that MNG increases early-layer update magnitude and noise responsiveness.

Table 7: Generation performance of GAT-XL/2 on ImageNet at different resolutions. Training at $512\times512$ achieves a similar FID-50K to the $256\times256$ setting with fewer epochs.

| Resolution | Model | Epochs | FID-50K |
|---|---|---|---|
| $256\times256$ | GAT-XL/2 | 20 | 4.02 |
| $512\times512$ | GAT-XL/2 | 15 | 4.04 |

The proposed multi-level noise-perturbed guidance is designed to counteract this asymmetry. By injecting noise signals at multiple depths and supervising the corresponding intermediate outputs with different noise strengths, the generator is encouraged to respond meaningfully to these perturbations across its entire depth rather than relying predominantly on the final blocks. This induces a coarse-to-fine usage of layers: earlier stages are trained to capture global, noise-robust structure under strong perturbations, while later stages progressively refine fine details as the noise level decreases. Consequently, the generator is nudged away from degenerate solutions where early layers collapse, and toward a regime where representational responsibility is more evenly distributed across layers, which we show leads to improved utilization and stronger overall performance.

**Additional analysis on inactivity of early layers** To quantify the inactivity of early layers, we performed a block-ablation study, as summarized in Fig. 4b. For each transformer block, we remove the block, regenerate the corresponding images, and measure the resulting change using a perceptual distance metric. We observe that, without MNG, ablating early blocks causes only minor changes in the generated images, even though these layers are expected to encode global, structural information. This suggests that the early part of the network is underutilized.

We further analyze this phenomenon by measuring feature similarity during generation. To this end, we sample 64 images per class for all classes using GAT-S/2. Then, we compute (1) the feature similarity between the input and output of each block, and (2) the similarity across different input noises $z$ (Fig. 9a and Fig. 9b). Similarity is defined as the cosine similarity at corresponding spatial locations, averaged over all tokens and images. For the model without MNG, both similarities are very high in the early layers (often close to 1), indicating that these blocks (i) induce only small changes in the features and (ii) are weakly influenced by the input noise. This is consistent with early-layer inactivity. In contrast, with MNG the similarities in early blocks are noticeably reduced for both measures, indicating that MNG increases the amount of feature change in early layers and makes them more responsive to the input noise.

## A.8 EXPERIMENTS ON IMAGENET-512

We verify that our model scales favorably to higher resolutions by training GAT-XL/2 on ImageNet at 512×512 resolution (7). We train for 15 epochs and obtain an FID-50K of 4.04, which is comparable to the 20 epochs result at 256×256 resolution. This suggests that our framework can achieve

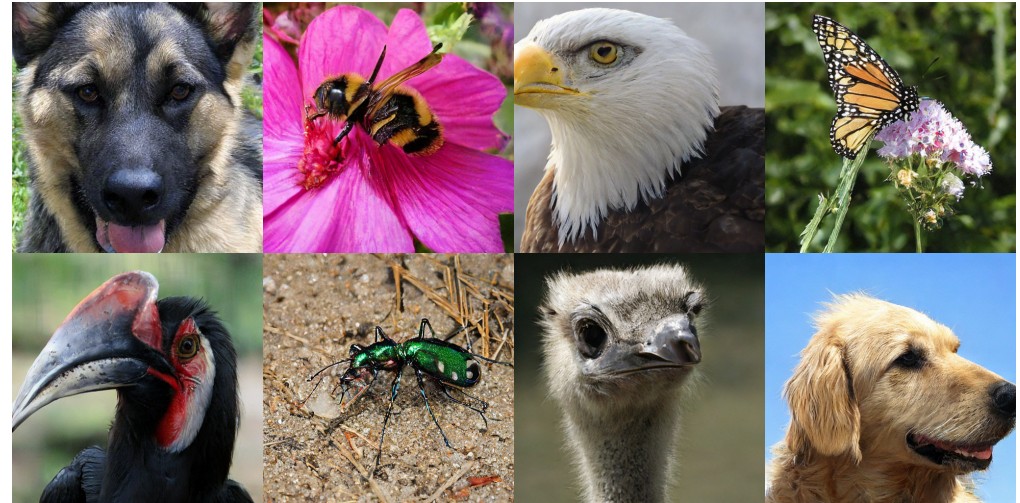

Figure 10: Examples from GAT-XL/2 on ImageNet-512.

Table 8: Text-to-image generation on MS-COCO at 256 resolution. We report FID and the number of function evaluations (NFE) at sampling time (lower is better). Methods marked with * use a CLIP image encoder.

| Method | Type | NFE | FID |
|---|---|---|---|
| Frido (Fan et al., 2023) | Diffusion | 200 | 8.97 |
| VQ-Diffusion (Gu et al., 2022) | Discrete diffusion | 100 | 19.75 |
| U-Net (Bao et al., 2023) | Diffusion | 50 | 7.32 |
| U-ViT-S/2 (Bao et al., 2023) | Diffusion | 50 | 5.95 |
| U-ViT-S/2 (Deep) (Bao et al., 2023) | Diffusion | 50 | 5.4 |
| AttnGAN (Xu et al., 2018) | GAN | 1 | 35.49 |
| DM-GAN (Zhu et al., 2019) | GAN | 1 | 32.64 |
| DF-GAN (Tao et al., 2022) | GAN | 1 | 19.32 |
| XMC-GAN (Zhang et al., 2021) | GAN | 1 | 9.33 |
| LAFITE* (Zhou et al., 2022) | GAN | 1 | 8.12 |
| MM-GAT (Ours) | GAN | 1 | 7.98 |

similarly strong performance with fewer epochs even at higher resolutions. Qualitative samples at 512×512 are provided in Fig. 10.

## A.9 EXPERIMENTS ON TEXT-TO-IMAGE GENERATION (MS-COCO)

We further evaluate our framework on text-to-image generation using MS-COCO (Lin et al., 2014) at $256^2$ resolution. Following the U-ViT (Bao et al., 2023) setting (Bao et al., 2023), we train on the MS-COCO training split and report FID on the validation set. For text encoding, we use a frozen CLIP text encoder, and adopt an MM-DiT (Esser et al., 2024a)-style conditioning scheme where the generator additionally receives the CLIP (Radford et al., 2021) word tokens as input (we refer to this model as MM-GAT), while the CLIP sentence embedding (i.e., the $[eot]$ token) is used as a global conditioning signal. We set a hidden dimension of 768 and a model depth of 24.

As shown in Table 8, MM-GAT attains competitive performance: although it slightly underperforms the best U-ViT variants in FID, it outperforms prior GAN-based approaches with 1-NFE. In particular, MM-GAT achieves a lower FID than LAFITE (Zhou et al., 2022), despite not using a CLIP image encoder. We emphasize that this is a deliberately lightweight, first-pass extension of GAT to the text-conditional setting, and we expect that modest additional tuning of this design could further improve performance. We also show the generated examples in Fig. 11.

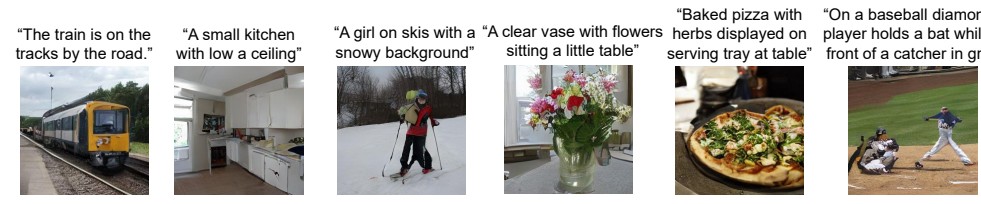

Figure 11: Examples of text-to-image generation on MS-COCO-256.

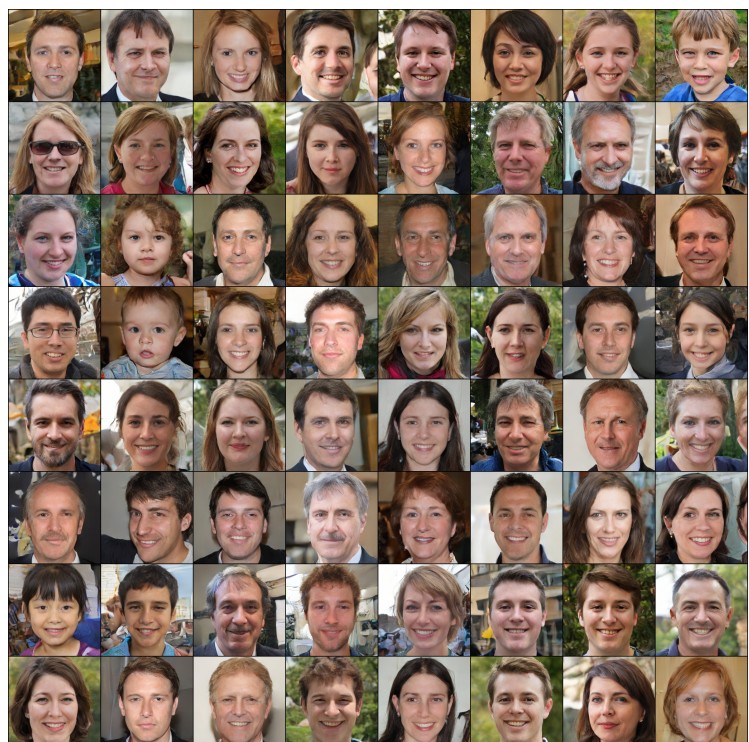

Figure 12: Unconditional generation examples from GAT-B/2 on FFHQ-256.

### A.10 Experiments on unconditional generation (FFHQ-256)

Our main experiments on ImageNet were designed to study how a transformer-based GAN scales in a complex, diverse, large-scale setting. To additionally verify the feasibility of unconditional training, we also train GAT-B/2 on FFHQ-256 (Fig. 12) for 25M images. When compared against a DiT-B/2 baseline trained under the identical size of generator and the same amount of data, our model achieves an FID of 9.74 versus 10.49 for DiT-B/2, indicating that GAT can successfully support unconditional generation and attains competitive performance with diffusion transformers even in unconditional generation.

### A.11 Intuition Behind the Width-Aware Learning Rate Rule

Eqn. 4 defines our width-aware learning rate schedule $\eta(C)$, which is designed to keep the *functional* update of the network approximately invariant as we change the channel dimension $C$. Here we provide additional intuition for this choice without a full mathematical proof.

Throughout the paper, when we refer to the *speed of change* of a network $f_\theta$, we mean the typical change in its outputs after a single optimization step, for example

$$\mathbb{E}_x\left[\|f_{\theta_{t+1}}(x) - f_{\theta_t}(x)\|\right], \tag{11}$$

where the expectation is over training samples $x$. Intuitively, this quantity measures how aggressively the function implemented by the network is updated per step, as opposed to the raw magnitude of parameter updates.

We make the following simplifying assumption, which is standard in analyses of wide neural networks: a hidden vector can be modeled as

$$x \in \mathbb{R}^C, \qquad x_i \sim \mathcal{N}(0, 1) \text{ i.i.d.,} \tag{12}$$

where $C$ is the channel dimension and we treat the channels as approximately independent and unit-variance. In this regime, the squared norm of the activation vector satisfies

$$\|x\|^2 = \sum_{i=1}^{C} x_i^2 \approx C, \tag{13}$$

so the "energy" of a feature vector grows approximately linearly with width.

For intuition, consider a single linear layer with scalar output

$$f_\theta(x) = w^\top x. \tag{14}$$

If the loss gradient with respect to this scalar output is $g$, and the learning rate is $\eta$, then a single SGD step updates the weights as

$$w' = w - \eta g x. \tag{15}$$

Evaluating the updated layer on the *same* input $x$, the output changes by

$$f_{\theta'}(x) - f_\theta(x) = (w')^\top x - w^\top x \tag{16}$$

$$= (w - \eta g x)^\top x - w^\top x \tag{17}$$

$$= -\eta g \|x\|^2. \tag{18}$$

Under the standardized-channel assumption $\|x\|^2 \approx C$, the typical magnitude of this per-step output change scales roughly as

$$|f_{\theta'}(x) - f_\theta(x)| \; \propto \; \eta C. \tag{19}$$

In words, for a *fixed* learning rate $\eta$, wider networks (larger $C$) tend to change their outputs more per step, simply because their activations (and hence their effective updates) carry more energy. The same mechanism applies layer by layer, so the end-to-end change in $f_\theta(x)$ inherits a similar dependence on $C$.

To keep the functional update scale approximately stable when we vary the width, we therefore choose a learning rate that decreases inversely with the channel dimension, as in Eq. (4),

$$\eta(C) \propto \frac{1}{C}, \tag{20}$$

so that the product $\eta C$ remains roughly constant. This, in turn, stabilizes the per-step change of intermediate features and final outputs as we transition from smaller to larger GAN architectures.

### A.12 RELATION TO DEPTH AND BATCH-SIZE SCALING

Our width-aware learning-rate rule in Eqn. 4 is derived from the goal of keeping the per-step change in the network outputs approximately constant. This principle is not inherent to width alone and, in principle, can be extended into a more comprehensive scaling law that also accounts for depth and batch size.

For deep transformer-style residual networks, stacking $L$ blocks increases the cumulative effect of each update. Under the same "constant functional update" view, one could combine our width-based rule with an additional depth-dependent factor, e.g., a $\sqrt{L}^{-1}$-type correction, or equivalently, adjust Layerscale initialization as a function of $L$ so that the overall update magnitude of the network remains similar across depths. Along the batch axis, our rule can be composed with standard learning-rate scaling heuristics used for large-batch transformer training, such as linear scaling ($\eta \propto B$) or square-root scaling ($\eta \propto \sqrt{B}$), where $B$ denotes the batch size.

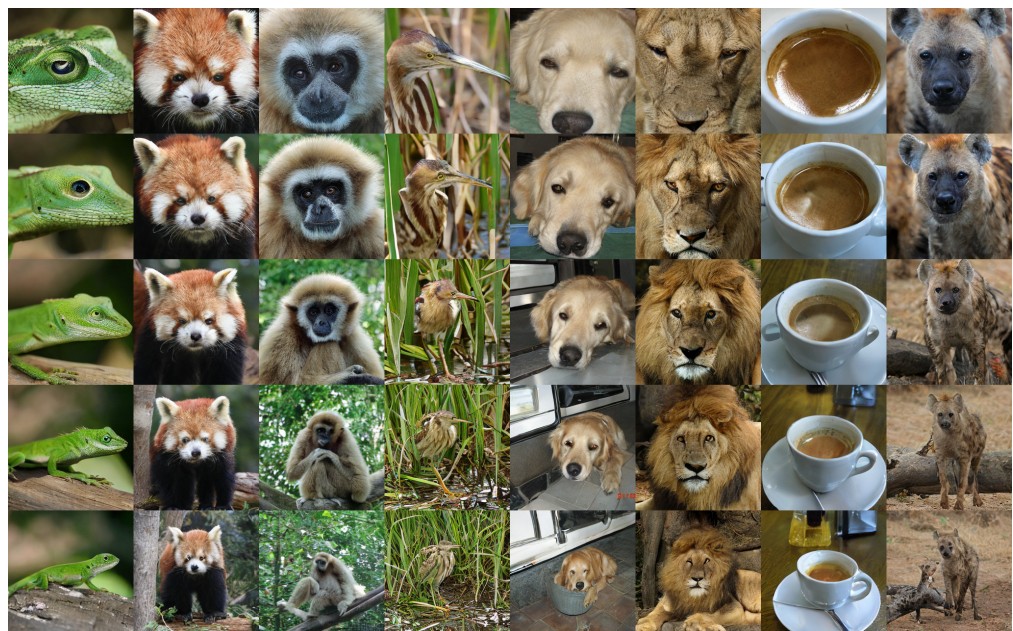

Figure 13: Image editing by GANSpace (Härkönen et al., 2020). We modify the top-2 principal components in the $W$ space, which produces a smooth zooming effect in the generated images.

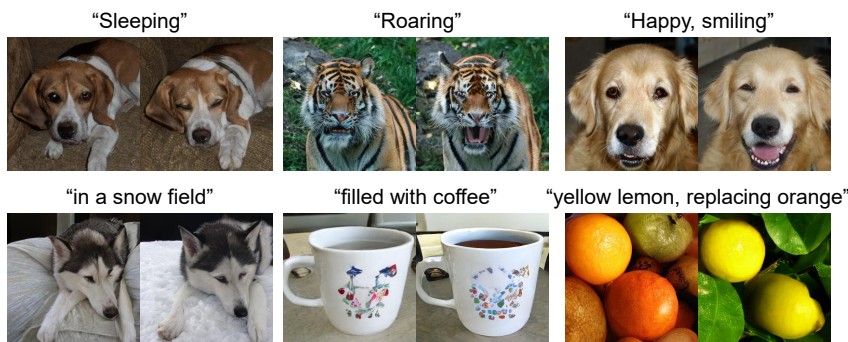

Figure 14: Image editing by StyleCLIP (Patashnik et al., 2021). The generated images faithfully follow the given text prompts, demonstrating that the edits successfully capture the desired semantics.

In this work, however, we intentionally focus on width for both conceptual clarity and empirical support. We conducted preliminary experiments in which we modified Layerscale initialization to compensate for depth, but within the depth range considered in our GAN architectures, this did not yield noticeable gains in stability or performance. Moreover, all of our main experiments use a fixed batch size of 512, so we do not yet have systematic evidence to justify incorporating batch-size dependence directly into the rule. Extending our width-aware learning-rate schedule to jointly cover width, depth, and batch size remains a promising direction for future work.

### A.13 IMAGE EDITING BY MANIPULATING THE LATENT SPACE OF GAT

To assess the transferability and robustness of the learned GAT latent space beyond unconditional sampling, we additionally evaluate the compatibility of the learned latent space with off-the-shelf editing methods. In particular, we apply GANSpace (Härkönen et al., 2020), which discovers unsupervised editing directions and manipulates the generation process along them, and Style-CLIP (Patashnik et al., 2021), which steers generated images to match a given text prompt. As illustrated in Fig. 13 and 14, both editing techniques transfer cleanly to GAT, producing smooth and

semantically meaningful variations, indicating that the learned latent space supports robust, reusable controls rather than overfitting to a single generative task.

### A.14 QUALITATIVE COMPARISON WITH OTHER METHODS

For qualitative comparison against strong one-step generative baselines, we evaluate MeanFlow and StyleGAN-XL on ImageNet-256. For MeanFlow, we use the PyTorch implementation and publicly released checkpoint[1] , which achieves a slightly better FID than reported in the original paper (3.39 FID compared to originally reported 3.43 FID (Geng et al., 2025)). As shown in Fig. 15-20, our method produces samples with noticeably higher fidelity than both baselines. For a fair comparison, we use a truncation value of 0.3 for both our model and StyleGAN-XL/2, while MeanFlow is trained by a guided flow field.

---

[1]https://github.com/zhuyu-cs/MeanFlow

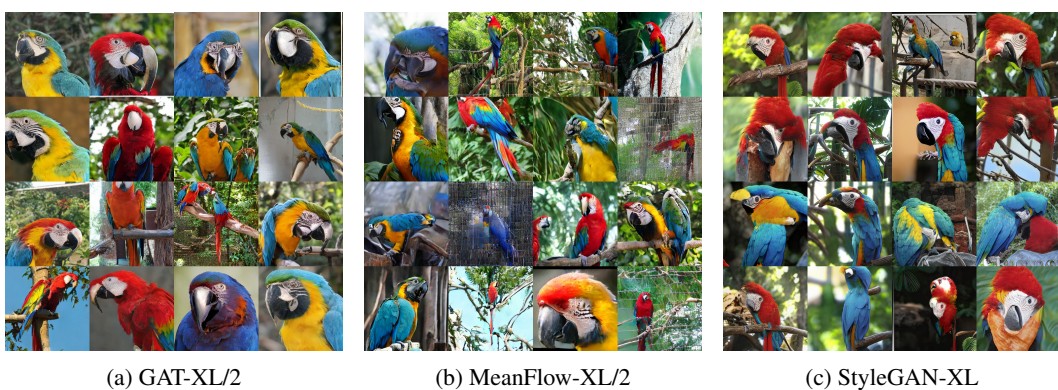

| (a) GAT-XL/2 | (b) MeanFlow-XL/2 | (c) StyleGAN-XL |

Figure 15: Qualitative comparison on ImageNet-256 by uncurated examples with 1-step generative models (MeanFlow (Geng et al., 2025) and StyleGAN-XL (Sauer et al., 2022), Class 88).

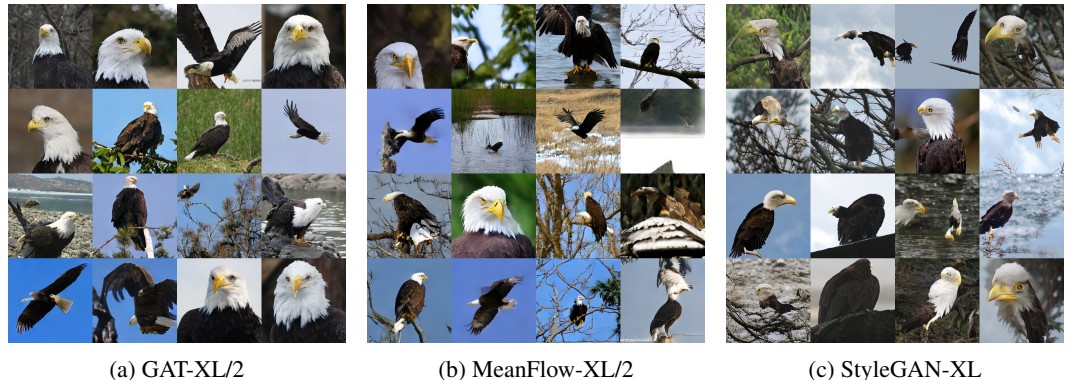

| (a) GAT-XL/2 | (b) MeanFlow-XL/2 | (c) StyleGAN-XL |

Figure 16: Qualitative comparison on ImageNet-256 by uncurated examples with 1-step generative models (MeanFlow (Geng et al., 2025) and StyleGAN-XL (Sauer et al., 2022), Class 22).

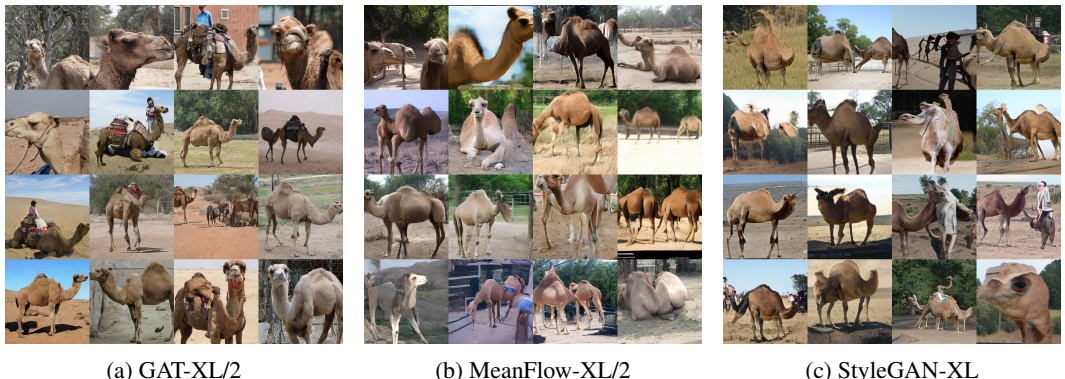

| (a) GAT-XL/2 | (b) MeanFlow-XL/2 | (c) StyleGAN-XL |

Figure 17: Qualitative comparison on ImageNet-256 by uncurated examples with 1-step generative models (MeanFlow (Geng et al., 2025) and StyleGAN-XL (Sauer et al., 2022), Class 354).

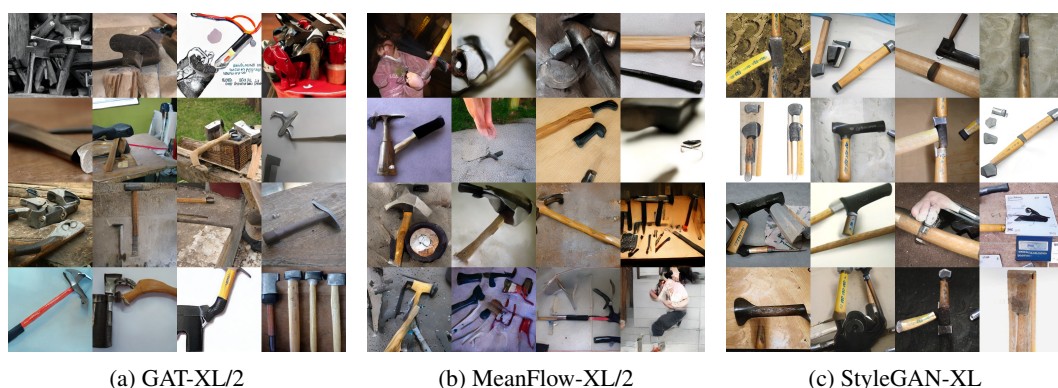

|          (a) GAT-XL/2          |          (b) MeanFlow-XL/2          |          (c) StyleGAN-XL          |

Figure 18: Qualitative comparison on ImageNet-256 by uncurated examples with 1-step generative models (MeanFlow (Geng et al., 2025) and StyleGAN-XL (Sauer et al., 2022), Class 587).

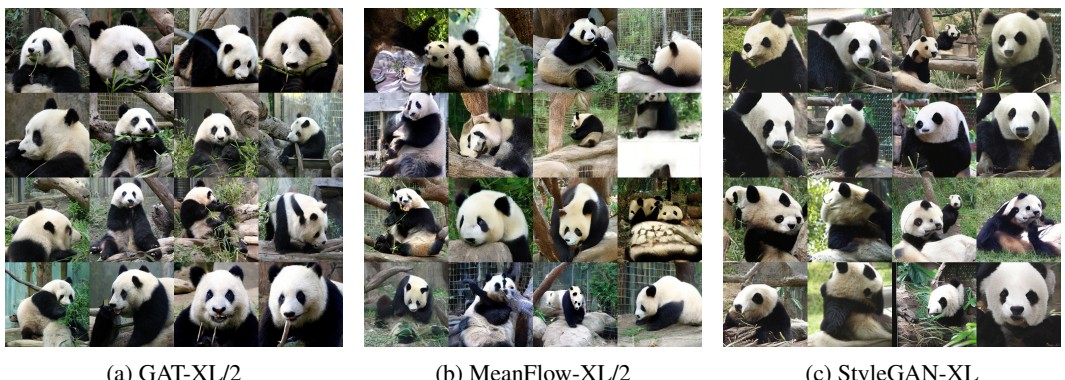

|          (a) GAT-XL/2          |          (b) MeanFlow-XL/2          |          (c) StyleGAN-XL          |

Figure 19: Qualitative comparison on ImageNet-256 by uncurated examples with 1-step generative models (MeanFlow (Geng et al., 2025) and StyleGAN-XL (Sauer et al., 2022), Class 388).

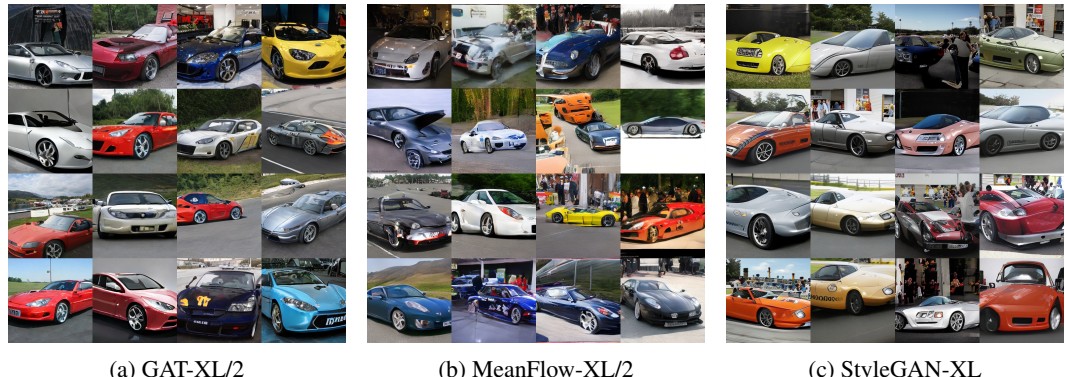

|          (a) GAT-XL/2          |          (b) MeanFlow-XL/2          |          (c) StyleGAN-XL          |

Figure 20: Qualitative comparison on ImageNet-256 by uncurated examples with 1-step generative models (MeanFlow (Geng et al., 2025) and StyleGAN-XL (Sauer et al., 2022), Class 817).

Table 9: Robustness of GAT to different image tokenizers on ImageNet-256. Both tokenizers use a downsampling ratio of 8.

| Tokenizer | Latent dim | Model | Epochs | FID-50K |
|---|---|---|---|---|
| SD-VAE | 4 | GAT-L/2 | 20 | 4.60 |
| FLUX-e2e (Leng et al., 2025) | 16 | GAT-L/2 | 20 | 3.73 |

### A.15 EXPERIMENTS ON A DIFFERENT TOKENIZER

We further assess the robustness of our framework to the choice of image tokenizer by training GAT on latents produced by an alternative encoder. Specifically, we encode ImageNet-256 using the recent FLUX-e2e tokenizer (Leng et al., 2025), which produces 16-dimensional latents with the same downsampling ratio of 8 as SD-VAE. In this latent space, we train a GAT-L/2 model for 20 epochs and obtain an FID-50K of 3.73, which surpasses the performance achieved with SD-VAE latents after 20 epochs of training (Table. 9). This result indicates that the proposed GAT framework is robust to the tokenizer choice and can potentially benefit even further from advances in image tokenizers.

### A.16 FORMAL SCALING LAW

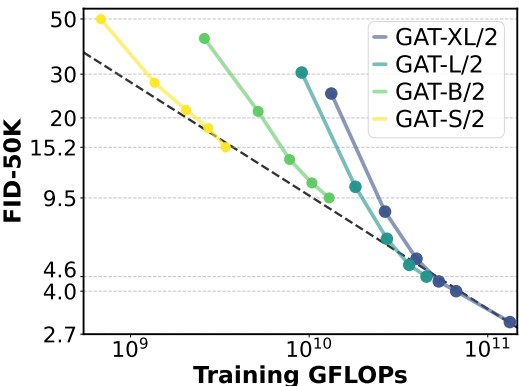

Figure 21: Compute–FID scaling of GAT models. We plot all training checkpoints of GAT-S/2, B/2, L/2, and XL/2 on a log–log plane, with total training compute $C$ (GFLOPs) on the x-axis and FID-50K on the y-axis. The dashed line shows a power-law fit over one summary point per model, following $\text{FID}(C) \approx 3.52 \times 10^5 \cdot C^{-0.456}$.

Fig. 21 analyzes the scaling behavior of our models with respect to training compute. We plot all training checkpoints for each model as trajectories on a log–log compute–FID plane, where the x-axis denotes the total training compute $C$ in GFLOPs (FLOPs per iteration × number of iterations with batch size 512, including G, D, VAE decoding, and the approximated GP), and the y-axis reports FID-50K. On top of these trajectories, we fit a power law using a single summary point per model, namely the final-iteration FID of GAT-S/2, B/2, L/2, and XL/2 (for GAT-XL/2 we use the 100K-iteration checkpoint reported in this paper and others for 50K-iteration). This yields the empirical relation

$$\text{FID}(C) \approx 3.52 \times 10^5 \cdot C^{-0.456}, \tag{21}$$

indicating a smooth, approximately power-law improvement of FID with training compute, consistent with scaling trends observed in diffusion and autoregressive models.

### A.17 ROBUSTNESS ACROSS RANDOM SEEDS

Due to the high computational cost, it was challenging to run exhaustive multi-seed experiments for all configurations. To get a rough sense of seed sensitivity, we trained GAT-S/2 on ImageNet-256 for 10 epochs (25K iterations) with three different random seeds. The resulting FID-5K scores

Table 10: Effect of VFM alignment (REPA) on FID-5K for different model sizes.

| Model | Epochs | REPA | FID-5K |
|---|---|---|---|
| GAT-S/2 | 10 | | 51.43 |
| GAT-S/2 | 10 | ✓ | 30.09 |
| GAT-S/2 | 20 | | 38.99 |
| GAT-S/2 | 20 | ✓ | 22.08 |
| GAT-B/2 | 10 | | 35.67 |
| GAT-B/2 | 10 | ✓ | 23.07 |

were 31.157, 28.907, and 31.069, compared to 30.085 for the originally reported run. These results suggest that the performance is not highly sensitive to the choice of random seed.

### A.18 DETAILED ANALYSIS OF VFM ALIGNMENT OBJECTIVE

**Computational overhead**   In this configuration (batch size 512, 4×RTX A6000 GPUs), computing the VFM alignment (REPA) term adds about 166 ms per iteration, corresponding to under 10% of the wall-clock time per training step even for the GAT-S/2 model. This overhead remains modest because the VFM encoder (DINOv2-B/16) is frozen and used only in forward mode (no backpropagation through the teacher), and the alignment is applied only to real images, not to generated samples. If this cost is still a concern, one can precompute and cache the teacher features for all real images before training, in which case the runtime cost of the VFM alignment during GAN training is almost negligible (only a lightweight projection and similarity computation remain).

**Effectiveness of VFM alignment across various model size**   We initially evaluated the effect of the VFM alignment objective (Eq. 9) through an ablation study on the GAT-S/2 model in the manuscript 5c. As summarized in Table 10, adding REPA consistently improves FID-5K: at 10 epochs, it reduces FID-5K from 51.43 to 30.09, and at 20 epochs, from 38.99 to 22.08. To verify that this effect is not tied to a particular model size, we additionally conduct an ablation on GAT-B/2. After training GAT-B/2 for 10 epochs without the VFM alignment term, the FID-5K degrades to 35.67, compared to 23.07 when the VFM alignment objective is used.

**Ablation of VFM alignment when fine-tuning the GAT**   We explicitly tested this scenario by fine-tuning a GAT-B/2 model after pretraining with VFM alignment. Starting from a 50K-iteration checkpoint trained with VFM alignment, we continued training for an additional 10K iterations without the VFM alignment term. In this setting, FID-5K increased slightly from 15.6 to 17.7, indicating that removing alignment does not cause catastrophic training collapse, but does lead to a moderate degradation in performance. We interpret this as the discriminator gradually losing the semantically meaningful features acquired during VFM alignment pretraining and thus providing weaker gradients, which is consistent with observations from self-supervised GANs (Chen et al., 2019) where discriminator features erode under prolonged adversarial training alone.

We expect that, even without VFM alignment, similar effects of losing features could be mitigated via alternative regularizers such as self-supervision (Oquab et al., 2023) or distilling generator features into the discriminator (e.g., GGDR-style objectives (Lee et al., 2022)), and we regard a systematic study of these alternatives as an interesting direction for future work.

### A.19 ADDITIONAL VISUALIZATIONS

In the following, we provide additional visualizations of our model. The section comprises parts as belows:

- Generated samples across model scales (20 epochs).
- Latent interpolation examples from GAT-XL/2.
- PCA visualizations of intermediate features from GAT-XL/2.
- Additional generation results from GAT-XL/2.

## A.20 GENERATED EXAMPLES FROM MODELS WITH VARIOUS SCALES (20 EPOCHS)

We provide uncurated examples generated from models with various scales. For fair comparison, we use the models trained for 20 epochs.

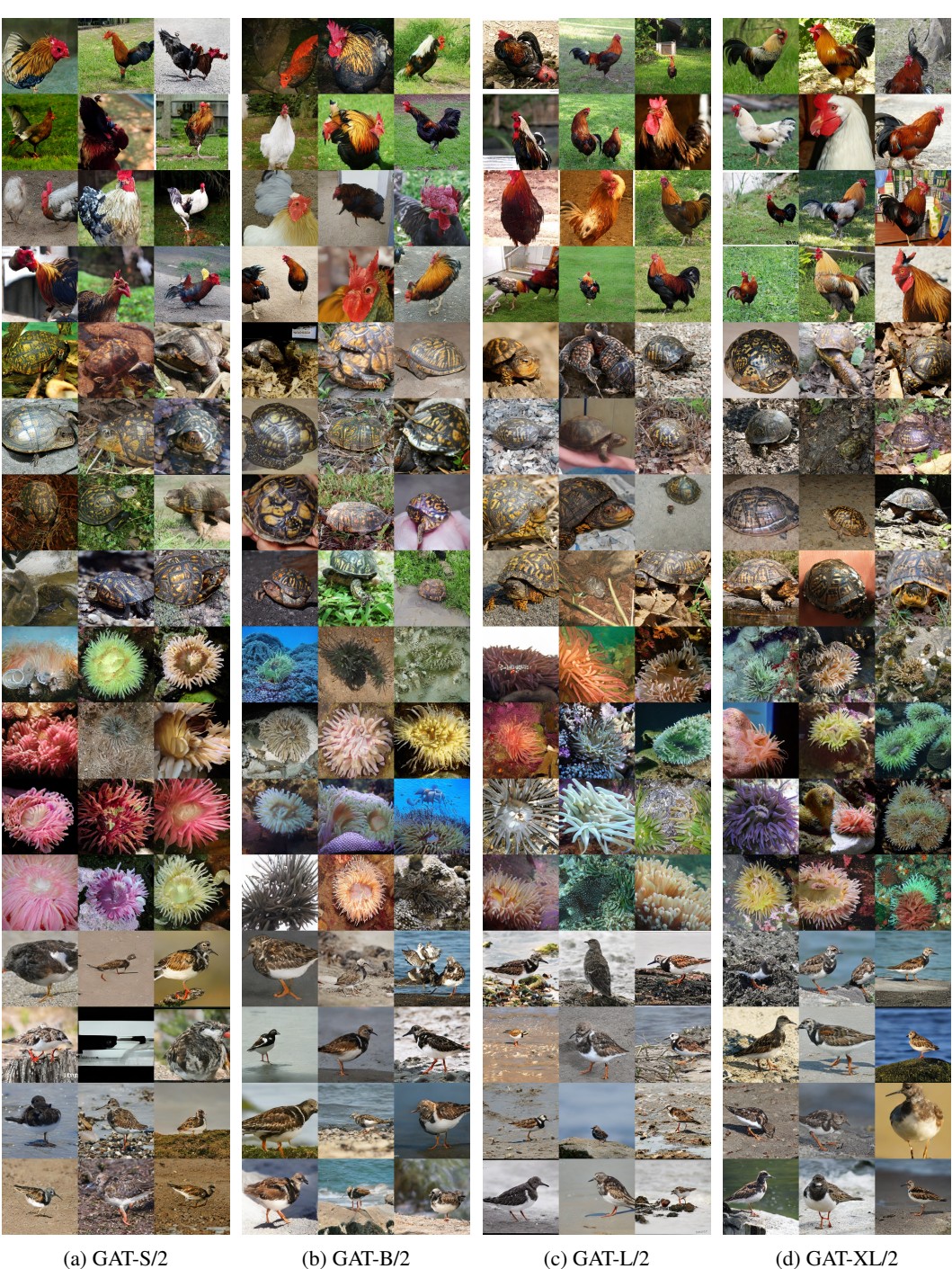

(a) GAT-S/2      (b) GAT-B/2      (c) GAT-L/2      (d) GAT-XL/2

Figure 22: Uncurated examples across model scales. From left to right, model size increases from GAT-S to GAT-XL. All models are trained for 50K iterations (i.e., 20 epochs).

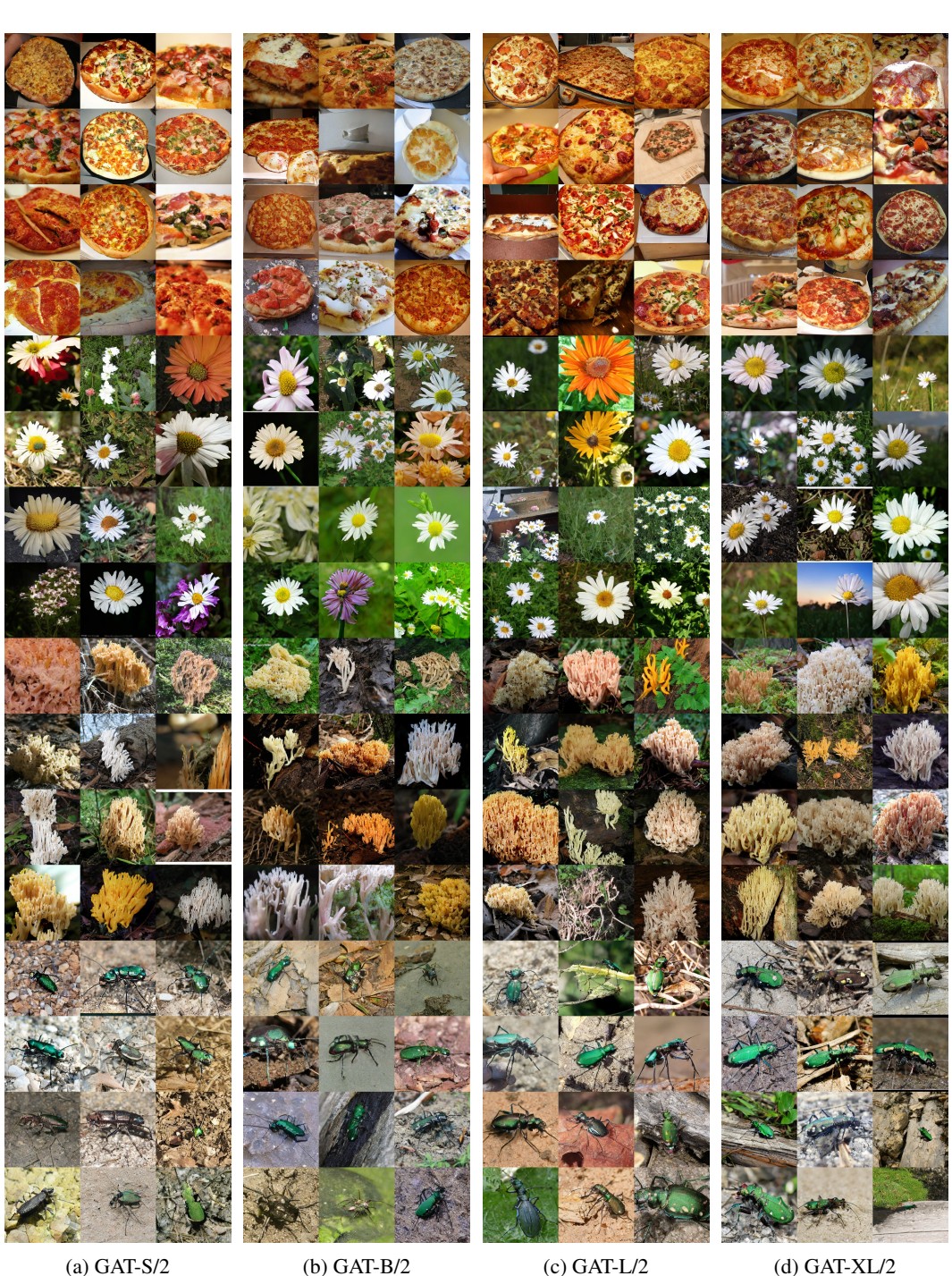

|  |  |  |  |
| --- | --- | --- | --- |
| (a) GAT-S/2 | (b) GAT-B/2 | (c) GAT-L/2 | (d) GAT-XL/2 |

Figure 23: Uncurated examples across model scales. From left to right, model size increases from GAT-S to GAT-XL. All models are trained for 50K iterations (i.e., 20 epochs).

A.21 LATENT INTERPOLATION EXAMPLES (GAT-XL/2)

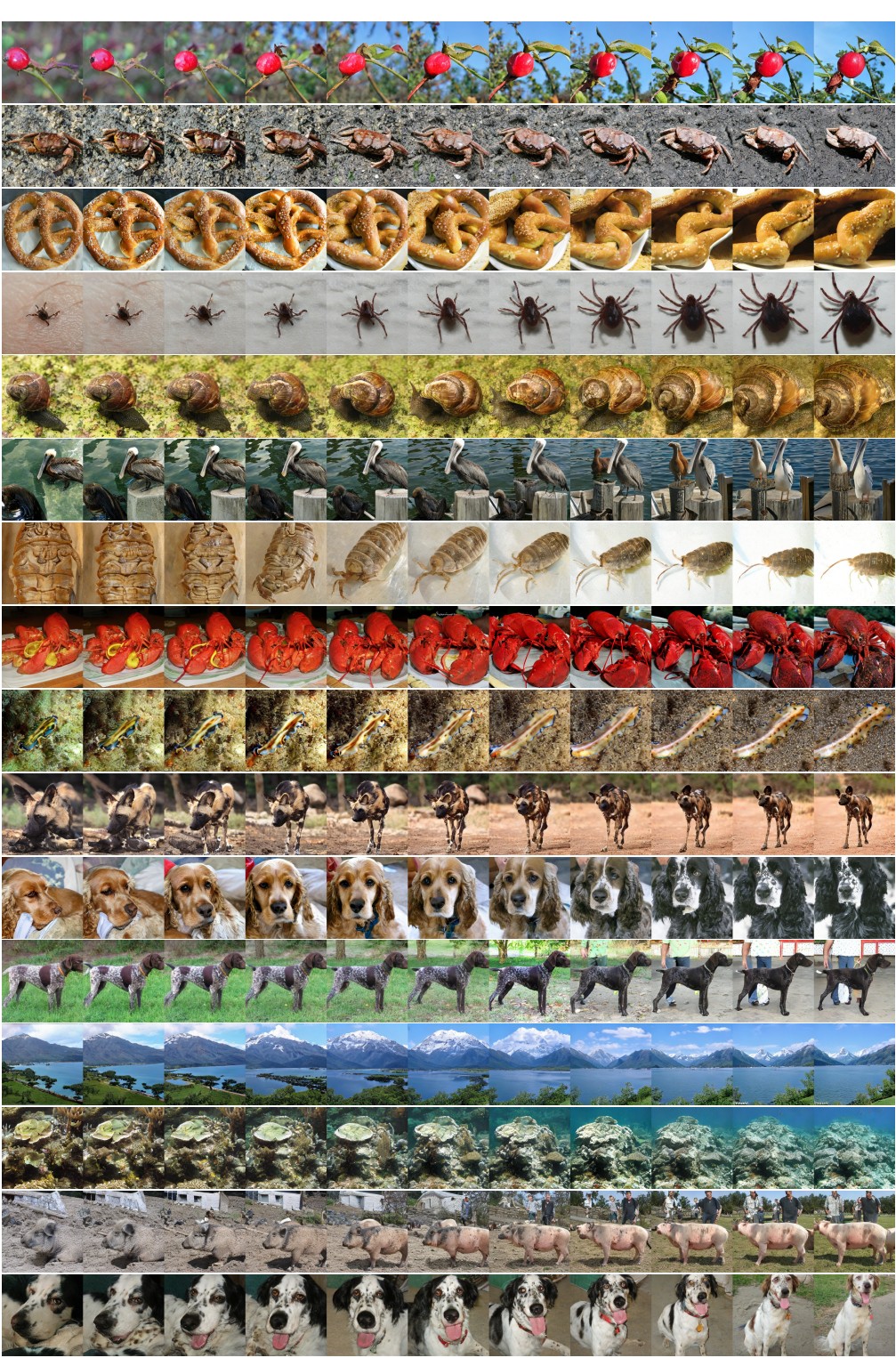

Figure 24: Latent interpolation examples between intra-class images.

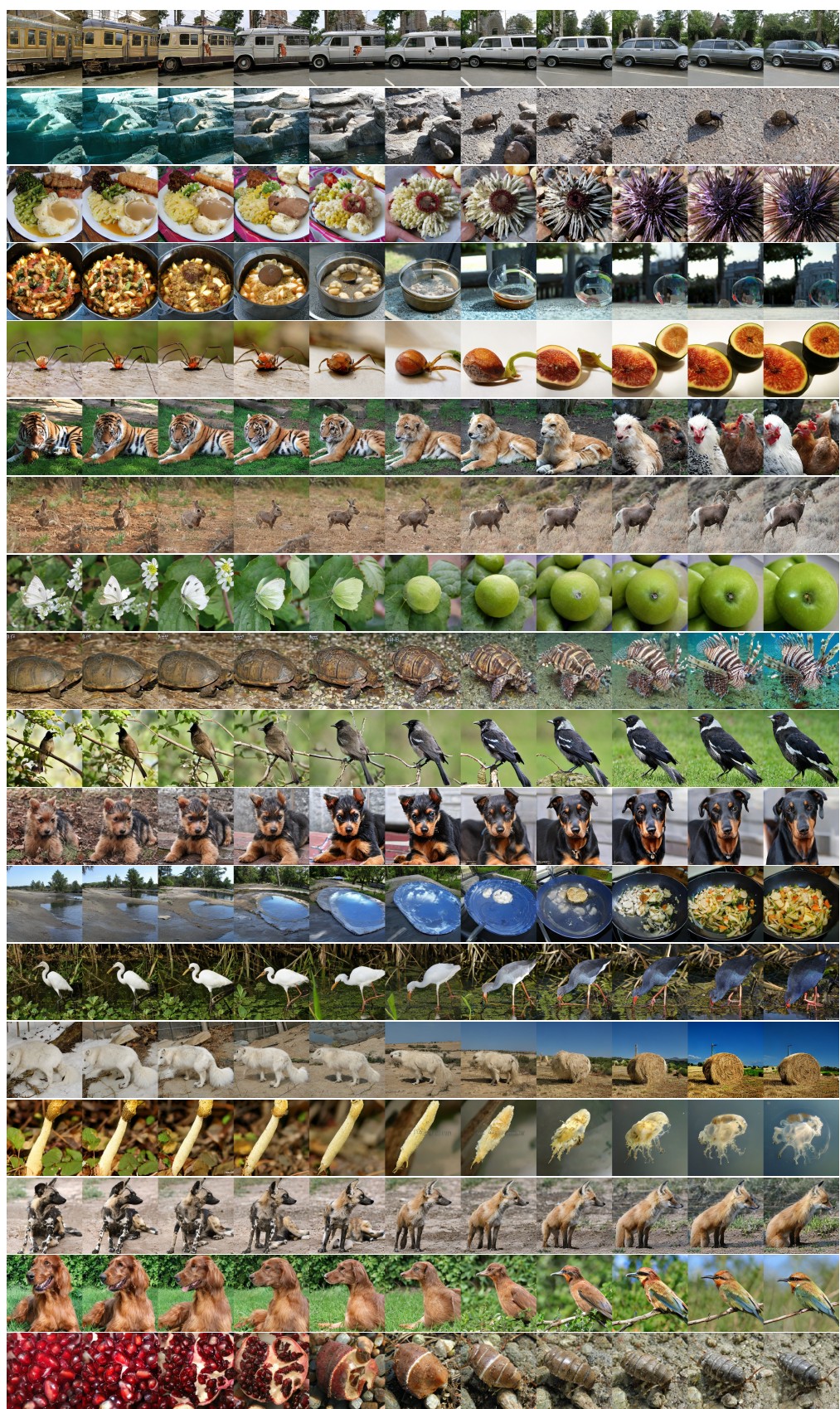

Figure 25: Latent interpolation examples between inter-class images.

## A.22 VISUALIZATION OF INTERMEDIATE FEATURES OF $G$ AND $D$ (GAT-XL/2)

We visualize intermediate features of $G$ and $D$ (GAT-XL/2) by projecting onto the top-3 PCA components. Visualizations are taken from every other block, with rows ordered as: image, feature, and attention map.

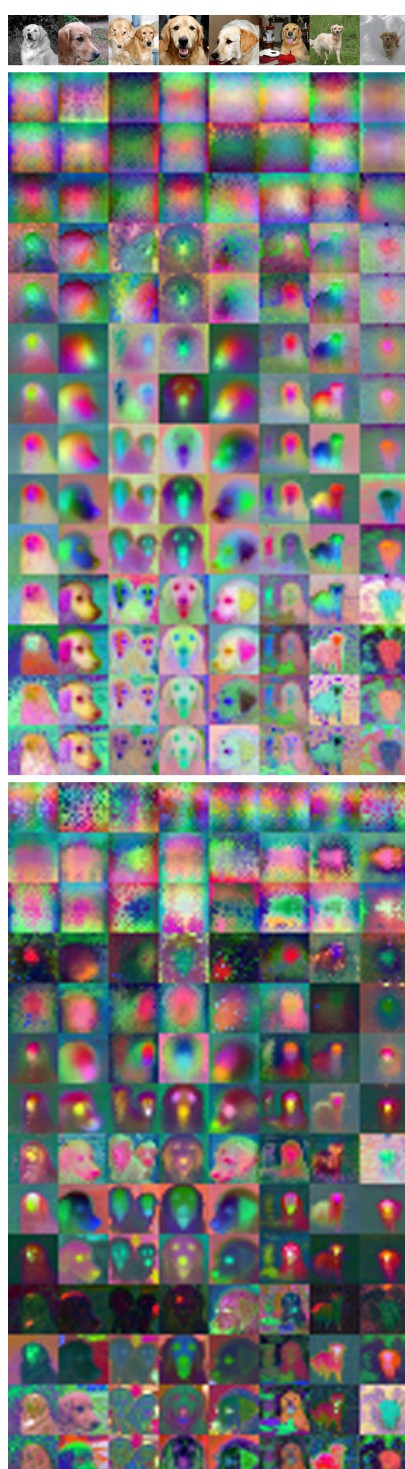

Figure 26: Feature visualization of $G$.

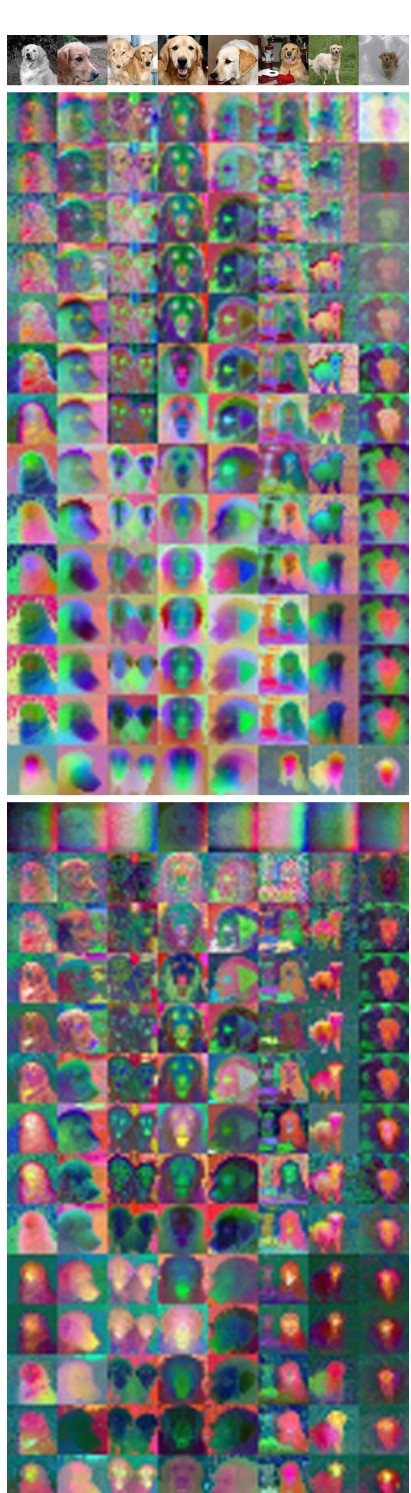

Figure 27: Feature visualization of $D$.

## A.23 ADDITIONAL QUALITATIVE EXAMPLES

Figure 28: Uncurated examples from GAT-XL/2 (40 epochs). Class 207, truncation $\psi$=0.85

Figure 29: Uncurated examples from GAT-XL/2 (40 epochs). Class 992, truncation $\psi$=0.85

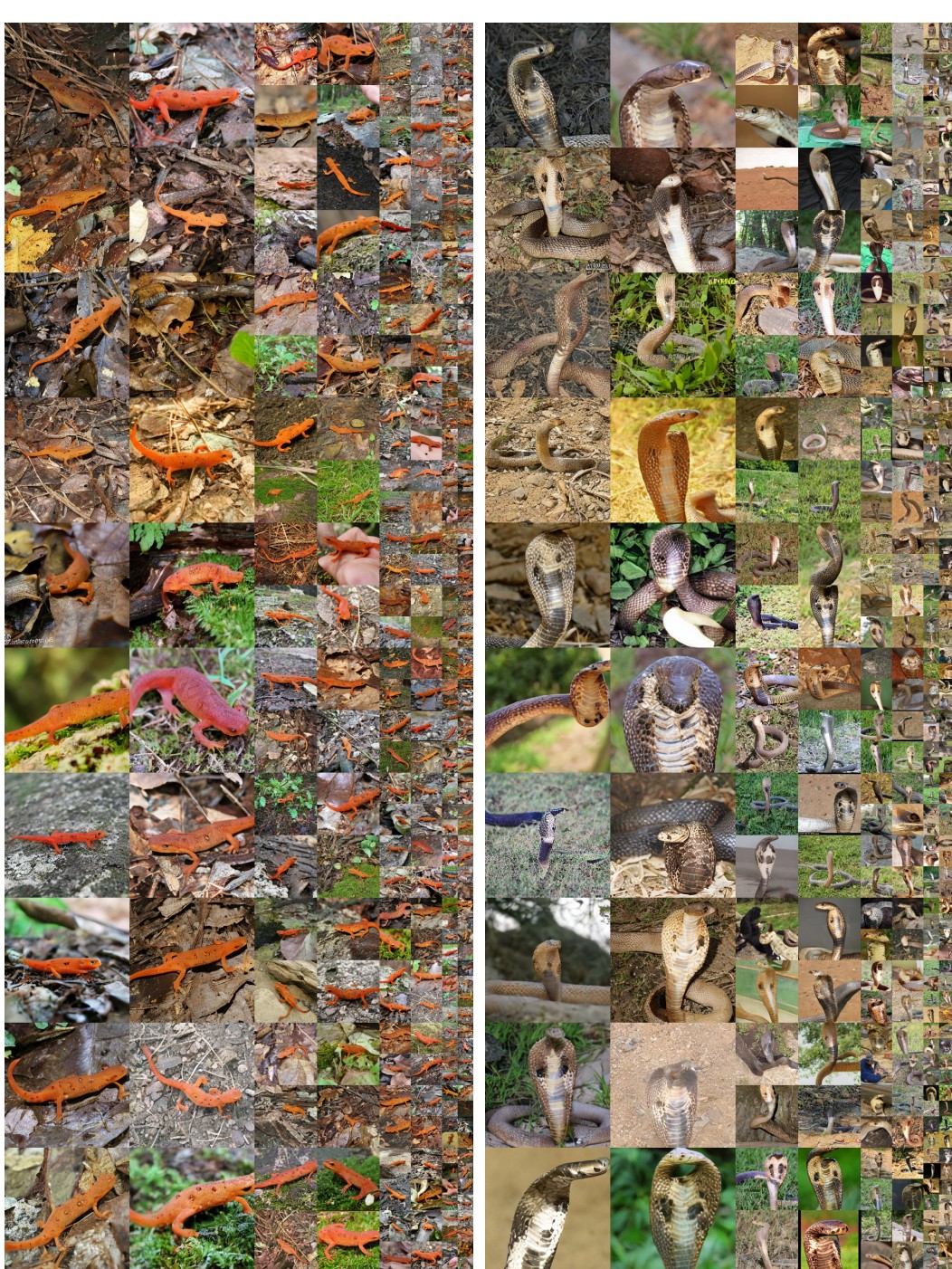

Figure 30: Uncurated examples from GAT-XL/2 (40 epochs). Class 27, truncation $\psi$=0.85

Figure 31: Uncurated examples from GAT-XL/2 (40 epochs). Class 63, truncation $\psi$=0.85

## A.24 USAGE OF LLM

We used an LLM as a writing assistant to help with the writing of the manuscript.

