# OpenReview forum: "Scalable GANs with Transformers"
_ICLR.cc/2026/Conference — Submitted to ICLR 2026_

### Official Review · Reviewer_dMUD · 2025-10-21

**Soundness:** 2
**Presentation:** 3
**Contribution:** 2
**Rating:** 4
**Confidence:** 4

**Summary:**

- The paper introduces Generative Adversarial Transformers (GAT) — a fully transformer-based GAN framework trained in a compact VAE latent space, enabling efficient and scalable adversarial training.

- It identifies two major issues when scaling GANs, inactive early generator layers and training instability. The authors solve them with (1) Multi-level Noise-perturbed Guidance (MNG) for early-layer activation and (2) a width-aware adaptive learning rate rule to stabilize training.

- GAT scales reliably from small to extra-large models, achieving state-of-the-art single-step ImageNet-256 generation with an FID of 2.96 in just 40 epochs — about 6× fewer than strong diffusion baselines.

- The work demonstrates that GANs can benefit from the same scalability principles as diffusion and autoregressive transformers, retaining GANs’ strengths (like single-step generation and latent manipulation) while achieving strong performance at large scale.

**Strengths:**

### 1. Technical Strengths

- Introduces a pure transformer-based GAN framework (GAT) that operates in VAE latent space, combining transformer scalability with GAN efficiency.

- Proposes two simple yet effective techniques: 1) Multi-level Noise-perturbed Guidance (MNG) for activating early generator layers and 2) Adaptive learning rate scaling to stabilize training across different model widths.

- Maintains a consistent transformer structure for both generator and discriminator without requiring architectural changes across scales.

- Incorporates representation alignment (REPA) between the discriminator and pre-trained Vision Foundation Models (DINOv2), enhancing discriminator features and stability.

### 2. Experimental Strengths

- Achieves state-of-the-art single-step image generation on ImageNet-256 with an FID of 2.96 in only 40 epochs, far fewer than diffusion/flow baselines.

- Demonstrates near-monotonic performance improvements (better FID) with increased model capacity, confirming successful scaling behavior.

- Includes detailed ablations for MNG, adaptive LR, and REPA, showing each component’s impact on stability and performance.

- Retains GANs’ key advantage, single-step inference, while outperforming many multi-step models in both speed and data efficiency.

**Weaknesses:**

### 1. Technical Limitations

- Relies on pre-trained VAE encoders (e.g., Stable Diffusion VAE), which may limit generality or introduce biases from the tokenizer.

- Transformer-based architectures can be memory-intensive, especially at large scales, limiting accessibility for lower-resource setups.

- Uses existing adversarial losses (R3GAN) with modest innovation; potential remains for more advanced loss formulations or self-distillation.

- While MNG alleviates inactive early layers, it adds auxiliary outputs and supervision complexity, which might scale non-linearly in deeper models.

### 2. Experimental Limitations

- Experiments are limited to ImageNet-256, leaving open questions about generalization to higher resolutions or more diverse domains (e.g., text-to-image).

- No perceptual or user-study validation of image quality, relies solely on FID metrics.

- Even though training is more efficient than diffusion models, the largest model (GAT-XL) still requires ~12 days on 8×A6000 GPUs.

- Does not test transferability or robustness of learned latent spaces for tasks like editing or cross-domain synthesis.

**Questions:**

See the weakness section. In addition:

- How sensitive is the GAT architecture to the choice of VAE latent space? Would performance change significantly with a different latent tokenizer?

- The Multi-level Noise-perturbed Guidance (MNG) adds auxiliary supervision. How does this affect training complexity or gradient flow compared to MSG-GAN (https://arxiv.org/abs/1903.06048)?

- Can you elaborate on how MNG interacts with adversarial training — is the discriminator jointly trained on all perturbed outputs or aggregated losses?

- How does your adaptive learning rate rule compare empirically to other scaling rules (e.g., linear or square-root scaling used in transformers)?

- Since both generator and discriminator are transformers, how do you handle instability due to self-attention depth or gradient explosion at large scales?

- Is the REPA alignment objective necessary for all scales, or does it mainly benefit large-capacity models?

- How much does the VFM alignment loss contribute to computational cost during training?

- Does the model’s performance degrade if the VFM alignment is removed after pretraining (e.g., fine-tuning without REPA)?

- How well does GAT generalize to other data modalities (e.g., faces, artwork, text-conditional generation)?

- You claim scalability properties similar to diffusion and autoregressive models — could you provide a formal scaling law or empirical scaling curve?

- Why does transformer scalability translate better to GANs in the latent space than in pixel space?

- Are there theoretical reasons why early generator layers become inactive without MNG?

- Why did you choose ImageNet-256 as the only benchmark? Have you evaluated GAT on higher resolutions or other datasets (e.g., FFHQ, COCO)?

- Could you provide qualitative examples comparing GAT outputs to StyleGAN-XL or DiT at similar FID levels?

- How consistent are your results across random seeds? what is the variance of FID across runs?

- What is the computational cost comparison (in GPU-days) between GAT-XL/2 and diffusion models with comparable FID?

- Did you observe any mode collapse or overfitting patterns as model capacity increased?

- How much improvement does latent-space training provide over pixel-space training, quantitatively?

- How does GAT compare in memory footprint and inference latency against diffusion models at similar quality?

- Does the use of VAE latent space constrain the diversity of generated samples (e.g., limits on fine texture detail)?

- How transferable are your findings to conditional or text-driven GANs (e.g., T2I models like GigaGAN)?

---

> ### Author Response · Authors · 2025-11-28
>
> Thank you for your valuable comments.
>
> ---
>
> ### 1. Generalization to higher resolutions or more diverse domains (e.g., text-to-image).
>
> We agree that restricting the main experiments to ImageNet-256 leaves open questions about how well GAT generalizes to higher resolutions and other domains such as text-conditioned generation. Our primary goal in this work was to study scaling along the model-capacity axis (parameters, width, depth) under a controlled resolution setting, which is why the core sweeps were conducted at 256×256.
>
> To directly address the reviewer’s concern, we have added two sets of experiments that go beyond the original ImageNet-256 setup. First, we extend GAT to a text-to-image setting on MS-COCO at 256×256 resolution by constructing a simple MM-GAT variant conditioned on CLIP text embeddings. Second, we train GAT-XL/2 on ImageNet-512, demonstrating that the same framework attains comparable FID-50K with fewer epochs at a substantially higher resolution. These results indicate that GAT is not tied to a single resolution or domain and can be naturally applied to both text-conditioned generation and higher-resolution image synthesis.
>
>
> **Text-to-image generation**
>
> In response to the reviewer’s comment, we added a text-to-image experiment on MS-COCO at 256 resolution.
>
> To this end, we build MM-GAT as follows. MM-GAT conditions GAT on CLIP text features using a simple scheme where the CLIP word tokens are fed into the generator (concatenated with the input positional encoding) and the [eot] embedding is used as a conditional vector, following an MM-DiT–style setup. Even with this simple conditioning mechanism and only minimal tuning under the short rebuttal timeline, the resulting MM-GAT already achieves competitive FID and outperforms prior 1-NFE GAN baselines (Table below).
>
> This indicates that GAT naturally extends to text-conditioned generation, and we expect that more careful tuning and refined conditioning designs could further improve performance.
>
> - **Text-to-image generation on MS-COCO at 256 resolution. We report FID and the number of function evaluations (NFE) at sampling time (lower is better). Ours achieves comparable FID with a lower NFE compared to diffusion models.**
> | Method           | Type               | NFE | FID   |
> |------------------|--------------------|-----|-------|
> | Diffusion models |                    |     |       |
> | VQ-Diffusion     | Discrete diffusion | 100 | 19.75 |
> | Frido            | Diffusion          | 200 | 8.97  |
> | U-Net            | Diffusion          | 50  | 7.32  |
> | U-ViT-S/2        | Diffusion          | 50  | 5.95  |
> | U-ViT-S/2 (Deep) | Diffusion          | 50  | 5.40  |
> | GAN models       |                    |     |       |
> | AttnGAN          | GAN                | 1   | 35.49 |
> | DM-GAN           | GAN                | 1   | 32.64 |
> | DF-GAN           | GAN                | 1   | 19.32 |
> | XMC-GAN          | GAN                | 1   | 9.33  |
> | LAFITE*          | GAN                | 1   | 8.12  |
> | MM-GAT (Ours)    | GAN                | 1   | 7.98  |
>
> Details and additional samples are provided in the Appendix A.9.
>
>
> **High-resolution generation (ImageNet-512)**
>
> In response to the reviewer’s request for higher-resolution results, we additionally train GAT-XL/2 on ImageNet at 512×512 resolution (Table below). With 15 epochs of training, we obtain an FID-50K of 4.04, which is comparable to the 4.02 FID-50K achieved at 256×256 resolution after 20 epochs. This suggests that our framework scales favorably to higher resolutions and can reach similarly strong performance with fewer epochs. Examples of 512×512 images are shown in Fig.10 in Appendix.
>
> - **Generation performance of GAT-XL/2 on ImageNet at different resolutions. Training at 512$\times$512 achieves a similar FID-50K to the 256$\times$256 setting with fewer epochs.**
> | Resolution  | Model    | Epochs | FID-50K |
> |------------|----------|--------|---------|
> | 256×256    | GAT-XL/2 | 20     | 4.02    |
> | 512×512    | GAT-XL/2 | 15     | 4.04    |
>
> Details and additional samples are provided in the Appendix A.8.
>
> ---

---

> ### Author Response · Authors · 2025-11-28
>
> ### 2. Concern about the high computational cost (training and inference cost)
>
> **Training cost**
>
> We agree that the largest GAT configuration is computationally demanding, mainly because, unlike a purely generative diffusion model, it jointly updates both a transformer generator and a strong discriminator at every iteration. However, high per-iteration cost does not directly translate to poor time-to-quality. For example, DiT-XL/2 on ImageNet-256 requires about 1400 epochs (36 days on 8×A100 GPUs) to reach FID 2.27, whereas our GAT-XL/2 attains ***a better FID-50K of 2.18 after only 60 epochs*** (18 days on 8×RTX A6000 GPUs). Thus, for comparable or better image quality, GAT actually needs fewer total passes over the data and substantially less wall-clock time than a diffusion transformer.
>
> We also note that most of this cost is dominated by the discriminator, which repeatedly evaluates real/fake samples and the gradient-penalty term. As discussed in the paper, we expect that using a pretrained feature extractor as the discriminator backbone (in the spirit of projected GANs) and fine-tuning only the later layers could significantly reduce this overhead, and systematically exploring such pretrained discriminators for GAT is an important direction for future work.
>
> **Inference cost**
>
> Regarding the runtime and memory comparison between GAT and diffusion transformers, GAT shares almost the same pure transformer backbone as DiT, so the per-step (per-NFE) cost and memory at matched width/depth are essentially similar. The main difference comes from the number of function evaluations: DiT typically requires around 250 denoising steps, whereas GAT is a single-step generator (1 NFE).
>
> In practice, this leads to about two orders of magnitude speedup. On a Titan RTX, when generating 64 ImageNet-256 samples (Table below), DiT-XL/2 with a 250-step sampler takes 15.26 seconds per image with 4.5 GB peak VRAM, while GAT-XL/2 takes only 0.077 seconds per image (≈200× faster) with 6.0 GB peak VRAM. The slightly higher VRAM for GAT is mainly due to caching intermediate features for the MNG guidance head; the transformer backbone itself has a comparable memory footprint to DiT.
>
>
> - **Inference latency and memory footprint on a single Titan RTX GPU (64 samples).**
> | Model           | NFE | Time / image (s) | Peak memory (MB) |
> |-----------------|-----|------------------|------------------|
> | DiT-XL/2        | 250 | 15.2612          | 4525.85          |
> | GAT-XL/2 (ours) | 1   | 0.0773           | 6028.11          |
>
> We include this discussion in A.6 in Appendix.
>
> ---
>
>
>
> ### 3. Concerns about Multi-level Noise Perturbed Guidance (MNG)
>
> **Scalability of MNG**
>
> The computational overhead of MNG is negligible in our implementation (auxiliary branches add only ≈0.015% extra FLOPs), so we do not expect it to become a bottleneck even for deeper models. Also, deeper generators may benefit more, as they have more early layers that would otherwise be weakly supervised. We also believe MNG itself can be further improved (e.g., number of scales, noise schedule), which we leave as future work.
>
> **How MNG interacts with adversarial training?**
>
> MNG interacts with adversarial training in a very lightweight way. We (1) attach small heads to a few intermediate generator features, (2) add fixed-strength noise to these head outputs, and (3) concatenate the results to the discriminator input in a single forward pass. The discriminator is thus trained jointly on all perturbed signals with no extra passes or separate loss terms. This only introduces a few shallow heads and slightly wider discriminator inputs, contributing ≈0.015% of the backbone FLOPs in GAT-XL/2.
>
> **Comparison with MSG-GAN**
>
> MNG and MSG-GAN are similar in that both feed multi-level generator features to the discriminator for auxiliary supervision. The key difference is how pairs are formed: MSG-GAN uses fixed downsampled versions of the same image, which can over-enforce cross-scale consistency and hurt fidelity, whereas MNG creates stochastically degraded counterparts by adding Gaussian noise at multiple levels, exposing D to a more diverse set of (image, degraded) pairs and yielding stronger intermediate supervision (see Fig. 3(a) for the MNG vs. MSG ablation). In terms of cost, MNG adds only negligible complexity, as noted above.
>
> **Theoretical reason**
>
> Our motivation comes from how GAN generators are supervised compared to other generative models. In diffusion and autoregressive models, each step is directly constrained by a reconstruction objective (e.g., denoising or next-token prediction), which enforces a strong, layer-wise coupling between the noise input and the final sample. In contrast, a standard GAN generator only receives supervision at the final image via the discriminator, so optimization can largely rely on the last few blocks, with much weaker pressure for early layers to carry meaningful structure.
>
> We include this discussion in Appendix A.7.
>
> ---

---

> ### Author Response · Authors · 2025-11-28
>
> ### 4. Concerns about Tokenizer (VAE)
>
> **Sensitivity of choice of tokenizer**
>
> To assess sensitivity to the latent space, we trained GAT on latents from an alternative tokenizer. Specifically, we re-encoded ImageNet-256 with the recent FLUX-e2e tokenizer [dMUD-1], which produces 16-dimensional latents with the same downsampling ratio (8×) as SD-VAE. In this space, a GAT-L/2 model trained for 13 epochs achieves an FID-50K of 4.58, which is comparable to the performance obtained with SD-VAE latents after 20 epochs. These results suggest that GAT is reasonably robust to the choice of tokenizer and can likely benefit further from future advances in image tokenizers.
>
> - **Robustness of GAT to different image tokenizers on ImageNet-256. Both tokenizers use a downsampling ratio of 8.**
> | Tokenizer            | Latent dim | Model   | Epochs | FID-50K |
> |----------------------|-----------:|--------:|-------:|--------:|
> | SD-VAE               | 4          | GAT-L/2 | 20     | 4.60    |
> | FLUX-e2e [dMUD-1]    | 16         | GAT-L/2 | 20     | 3.73    |
>
> We add this experiment in Appendix. A.15.
>
> [dMUD-1] Leng, Xingjian, et al. "Repa-e: Unlocking vae for end-to-end tuning with latent diffusion transformers." arXiv preprint arXiv:2504.10483 (2025).
>
>
> **Diversity of generated samples**
>
> We acknowledge that a VAE latent space is less expressive than the full image domain and that GAT inherits this limitation. However, most modern high-resolution generative models (diffusion and autoregressive) also operate in VAE latents and still achieve high visual quality and diversity, including fine textures. Our experiments similarly show that GAT attains comparable image quality while being trained purely in VAE latent space.
>
> **Improvement of latent-space training over pixel-space training**
>
> We mainly adopt VAE latents to place computation in an efficient low-resolution space where wide/deep transformers are trainable; a strictly comparable pixel-space version would require a different architecture (e.g., an additional high-capacity decoder), so any number we report would be heavily confounded. For this reason, we do not claim a precise quantitative “gain” over pixel-space training, and instead focus on scaling within latent space. We also note that most of our strongest baselines already operate in VAE latent space, so our comparisons are made in this standard tokenizer regime rather than against a much weaker pixel-space model.
>
> ---
>
> ### 5. Comparison of adaptive learning rate rule with other scaling rules
>
> Our adaptive learning-rate rule is complementary to the standard scaling rules used in large-batch transformer training, rather than a competing alternative. Conventional linear or square-root scaling rules focus on how the learning rate should change with batch size, motivated primarily by gradient statistics (e.g., variance reduction as the batch grows). In contrast, our rule is derived from a different perspective: we explicitly target keeping the per-step change in the network output approximately constant as we vary the width (and hence the number of parameters). These two views can therefore be composed: the batch-size–dependent rules control the stochasticity of the gradient, while our width-based rule controls the functional update of the network.
>
> We further believe that this principle can be extended toward a more comprehensive scaling law that also accounts for depth and batch size. In deep transformer-style residual networks, stacking L blocks effectively increases the cumulative step size, so one could combine our width-based rule with an additional $\sqrt(L)^{-1}$-type factor (or equivalently, depth-dependent Layerscale initialization) to preserve a similar overall update magnitude. Along the batch axis, our rule can naturally be used together with the standard linear or square-root learning-rate scaling heuristics commonly employed in large-batch training.
>
> However, in this work we intentionally focused on width for clarity and empirical support. We performed preliminary experiments where we adjusted Layerscale initialization to compensate for depth, but within our depth range this did not yield a noticeable improvement in stability or performance, so we did not further pursue depth-dependent tuning. In addition, all of our main experiments use a fixed batch size (512), so we do not yet have systematic evidence to justify building batch-size dependence directly into our rule. We therefore view extending the rule jointly over width, depth, and batch size—and empirically comparing such combined schemes with existing scaling rules—as a promising direction for future work.
>
> We include this discussion in A.12 in Appendix.
>
> ---

---

> ### Author Response · Authors · 2025-11-28
>
> ### 6. Training stability
>
> Despite the common belief that transformer-based GANs are unstable, we did not observe pronounced mode collapse or overfitting as we increased model capacity. We found that a modern GAN objective together with a carefully designed transformer architecture is sufficient to stabilize training. In particular, following R3GAN, using both R1 and R2 (real/fake zero-centered gradient penalties) had the largest positive impact. In addition, standard transformer design choices such as QK normalization and initializing the class embeddings and [CLS] token with a scale of 0.02 also contributed to stability. As a result, our largest models can be trained as pure transformers without any additional regularizers such as L2 attention or spectral normalization, while still achieving stable adversarial training across scales.
>
> ---
>
> ### 7. Concerns about VFM alignment (REPA)
>
> **Computational cost of REPA**
>
> The extra cost from the VFM alignment loss is practically negligible.
>
> On our setup (batch size 512, 4× RTX A6000), computing the VFM alignment (REPA) term adds about **166 ms per iteration**, corresponding to **well under 10% of the wall-clock time per training step**, even for the GAT-S/2 model. This overhead is modest because:
>
> - the VFM encoder (DINOv2-B/16) is **frozen** and used **only in forward mode** (no backprop through the teacher), and
> - the alignment is applied **only to real images**, not to generated samples.
>
> If this cost is still a concern, one can precompute and cache the teacher features for all real images once before training; in that case, the runtime overhead of the VFM alignment loss during GAN training effectively disappears (only a very small projection + similarity computation remains).
>
> We include this discussion in Appendix A.18.
>
> **Effectiveness of REPA across model size**
>
> In the original submission, we evaluated this component via an ablation on the GAT-S/2 model (Fig.5c). As summarized in Table below, adding REPA consistently improves FID-5K: at 10 epochs, the FID-5K is reduced from 51.43 to 30.09, and at 20 epochs, from 38.99 to 22.08.
>
> To address the concern that this effect might be specific to a particular model size, we additionally conduct an ablation on a larger model, GAT-B/2, in the revision. After training GAT-B/2 for 10 epochs without the VFM alignment term, the FID-5K degrades to 35.67, compared to 23.07 when the VFM alignment objective is used. These results suggest that the benefit of the VFM alignment objective is robust across different model scales.
>
> - **Effect of VFM alignment (REPA) on FID-5K for different model sizes.**
> | Model   | Epochs | REPA | FID-5K |
> |--------|--------|------|--------|
> | GAT-S/2 | 10    |      | 51.43 |
> | GAT-S/2 | 10    | ✓    | 30.09 |
> | GAT-S/2 | 20    |      | 38.99 |
> | GAT-S/2 | 20    | ✓    | 22.08 |
> | GAT-B/2 | 10    |      | 35.67 |
> | GAT-B/2 | 10    | ✓    | 23.07 |
>
> We include this discussion in Appendix A.18.
>
> **Concerns about degradation of model’s performance as removing REPA after pretraining**
>
> Regarding the reviewer’s question about what happens if VFM alignment is removed after pretraining, we explicitly tested this scenario by fine-tuning a GAT-B/2 model after pretraining with VFM alignment. Starting from a 50K-iteration checkpoint trained with VFM alignment, we continued training for an additional 10K iterations without the VFM alignment term. In this setting, FID-5K increased slightly from 15.6 to 17.7, indicating that removing alignment does not cause catastrophic training collapse, but does lead to a moderate degradation in performance.
>
> We interpret this behavior as the discriminator gradually losing the semantically meaningful features acquired during VFM alignment pretraining and thus providing weaker gradients, which is consistent with observations from self-supervised GANs[dMUD-2] where discriminator features erode under prolonged adversarial training alone. We also expect that, even without VFM alignment, similar feature erosion could be mitigated via alternative regularizers such as self-supervision or distilling generator features into the discriminator, and we view a systematic study of these alternatives as an interesting direction for future work.
>
> We include this discussion in Appendix A.18.
>
> [dMUD-2] Chen, Ting, et al. "Self-supervised gans via auxiliary rotation loss." Proceedings of the IEEE/CVF conference on computer vision and pattern recognition. 2019.
>
> ---

---

> ### Author Response · Authors · 2025-11-28
>
> ### 8. Formal or emprical scaling rule
>
> We agree that the scaling aspect was not explicit enough and will clarify it in the revision. In the new compute–FID plot, we show **all training checkpoints** (all iterations for each model) as points/curves on a log–log axis, where the x-axis is total training compute C in GFLOPs (FLOPs per iteration × number of iterations, batch size 512, including G, D, VAE decode, and approximated GP), and the y-axis is FID-50K.
>
> On top of these points, we add a **power-law fit** using **only one point per model**, namely the final-iteration FID of each GAT-S/2, B/2, L/2, and XL/2 (for GAT-XL/2 we use the 100K-iteration model reported in the paper). This yields the empirical relation
>
> $\text{FID}(C) \approx 3.52 \times 10^{5} \cdot C^{-0.456}$,
>
> i.e., a smooth, approximately power-law improvement of FID with training compute, in line with empirical scaling curves reported for diffusion and autoregressive models.
>
> We add this discussion and the plot about scaling rule in Appendix Fig.21.
>
>
> ---
>
> ### 9. Qualitative comparison with prior methods such as StyleGAN-XL and Diffusion models
>
> In response to the reviewer’s request for qualitative comparisons against strong one-step generative baselines, we additionally evaluated MeanFlow and StyleGAN-XL on ImageNet-256 and included side-by-side samples in the revised manuscript.
>
> As shown in Appendix Fig.15-Fig.20, our method produces samples with noticeably higher fidelity than both MeanFlow and StyleGAN-XL. For a fair comparison, we use a truncation value of 0.3 for both our model and StyleGAN-XL/2, while MeanFlow is evaluated with its guided flow field as in the original setup.
>
>
> ---
>
> ### 10. Robustness across random seeds
>
> Due to the high computational cost, it was challenging to run exhaustive multi-seed experiments for all configurations. To get a rough sense of seed sensitivity, we trained GAT-S/2 on ImageNet-256 for 10 epochs (25K iterations) with three different random seeds. The resulting FID-5K scores were 31.157, 28.907, and 31.069, compared to 30.085 for the originally reported run. These results suggest that the performance is not highly sensitive to the choice of random seed.
>
> We add this experiment in Appendix A.17.
>
>
> ---
>
> ### 11. Transferability of learned latent spaces (editing)
>
> Regarding the reviewer’s question about whether the learned GAT latent space is robust and transferable beyond unconditional sampling, we additionally evaluated its compatibility with off-the-shelf editing methods. Specifically, we applied GANSpace [dMUD-3], which discovers unsupervised editing directions and manipulates the generation process along those directions, and StyleCLIP [dMUD-4], which steers generated images toward a target text prompt.
>
> As shown in Appendix Figs.13 and Fig.14, both editing techniques transfer cleanly to GAT, yielding smooth and semantically meaningful variations. This suggests that the learned latent space provides robust, reusable control dimensions rather than overfitting to a single generative task.
>
> ---
>
>
>
> [dMUD-3] Härkönen, Erik, et al. "Ganspace: Discovering interpretable gan controls." Advances in neural information processing systems 33 (2020): 9841-9850.
>
> [dMUD-4] Patashnik, Or, et al. "Styleclip: Text-driven manipulation of stylegan imagery." Proceedings of the IEEE/CVF international conference on computer vision. 2021.
>
> ####

---

### Official Review · Reviewer_Y3NZ · 2025-10-31

**Soundness:** 3
**Presentation:** 3
**Contribution:** 2
**Rating:** 4
**Confidence:** 2

**Summary:**

This works looks to bring in scaling law perspectives for GANs, noting that there is scant literature on showing how to scale GANs to large settings. The authors motivate the need based on the fact that GAN inference is a single shot, as opposed to say diffusion, as well as interpretability in the form of latent interpolation. They use transformer architectures for the the generator and discriminator, and highlight two key observations in scaling: use of intermediate feedback (multi-level noise perturbed guidance) to ensure all layers of the generator contribute, and a scaling rule to adjust hyper parameters to ensure training stability.

While the proposed methodology is simple and well motivated, I am not particularly convinced by the experiments. For a paper on scaling, the image generated (the largest being 256 x 256) are not substantially large to demonstrate scaling effects. The paper also lacks a theoretical motive/analysis of the multi-level noise perturbed guidance. This is not a major issue, but mean I must focus primarily on the experiments.

That said, this is slightly outside my area of expertise and happy to engage with the other reviewers on this.

**Strengths:**

See above.

**Weaknesses:**

See above.

**Questions:**

Can you comment on the weakness.

---

> ### Author Response · Authors · 2025-11-28
>
> Thanks for your valuable comments.
>
> ---
>
> ### 1. Small resolution dataset to prove its scaling effect
>
> We agree that 256×256 is not “large” in an absolute sense for image generation. Our intent in this work, however, is to study **scaling with respect to model capacity (parameters, width, depth)** rather than primarily scaling image resolution.
>
> We will clarify this in the paper and also add supporting results:
>
> - We have additionally trained our framework at 512×512 resolution, and on another modality (a text-to-image setup).
>
> In the revised version, we will (i) explicitly state that our main scaling axis is model size, with resolution kept at 256×256 for controlled sweeps, and (ii) include 512×512 and text-to-image results in the appendix to demonstrate that the proposed GAT can be extended beyond the baseline settings.
>
>
> **Text-to-image generation**
>
> In response to the reviewer’s comment, we added a text-to-image experiment on MS-COCO at 256 resolution.
>
> To this end, we build MM-GAT as follows. MM-GAT conditions GAT on CLIP text features using a simple scheme where the CLIP word tokens are fed into the generator (concatenated with the input positional encoding) and the [eot] embedding is used as a conditional vector, following an MM-DiT [Y3NZ-1]–style setup. Even with this simple conditioning mechanism and only minimal tuning under the short rebuttal timeline, the resulting MM-GAT already achieves competitive FID and outperforms prior 1-NFE GAN baselines (Table below).
>
> This indicates that GAT naturally extends to text-conditioned generation, and we expect that more careful tuning and refined conditioning designs could further improve performance.
>
> - **Text-to-image generation on MS-COCO at 256 resolution. We report FID and the number of function evaluations (NFE) at sampling time (lower is better). Ours achieves comparable FID with a lower NFE compared to diffusion models.**
> | Method           | Type               | NFE | FID   |
> |------------------|--------------------|-----|-------|
> | Diffusion models |                    |     |       |
> | VQ-Diffusion     | Discrete diffusion | 100 | 19.75 |
> | Frido            | Diffusion          | 200 | 8.97  |
> | U-Net            | Diffusion          | 50  | 7.32  |
> | U-ViT-S/2        | Diffusion          | 50  | 5.95  |
> | U-ViT-S/2 (Deep) | Diffusion          | 50  | 5.40  |
> | GAN models       |                    |     |       |
> | AttnGAN          | GAN                | 1   | 35.49 |
> | DM-GAN           | GAN                | 1   | 32.64 |
> | DF-GAN           | GAN                | 1   | 19.32 |
> | XMC-GAN          | GAN                | 1   | 9.33  |
> | LAFITE*          | GAN                | 1   | 8.12  |
> | MM-GAT (Ours)    | GAN                | 1   | 7.98  |
>
> Details and additional samples are provided in the Appendix A.9.
>
> [Y3NZ-1] Esser, Patrick, et al. "Scaling rectified flow transformers for high-resolution image synthesis." Forty-first international conference on machine learning. 2024.
>
> **High-resolution generation (ImageNet-512)**
>
> In response to the reviewer’s request for higher-resolution results, we additionally train GAT-XL/2 on ImageNet at 512×512 resolution (Table below). With 15 epochs of training, we obtain an FID-50K of 4.04, which is comparable to the 4.02 FID-50K achieved at 256×256 resolution after 20 epochs. This suggests that our framework scales favorably to higher resolutions and can reach similarly strong performance with fewer epochs. Examples of 512×512 images are shown in Fig.10 in Appendix.
>
> - **Generation performance of GAT-XL/2 on ImageNet at different resolutions. Training at 512$\times$512 achieves a similar FID-50K to the 256$\times$256 setting with fewer epochs.**
>
> | Resolution  | Model    | Epochs | FID-50K |
> |------------|----------|--------|---------|
> | 256×256    | GAT-XL/2 | 20     | 4.02    |
> | 512×512    | GAT-XL/2 | 15     | 4.04    |
>
> Details and additional samples are provided in the Appendix A.8.
>
> ---

---

> > ### Author Response · Authors · 2025-11-28
> >
> > ### 2. Lack of motiovaiton/experiments about Multi-level Noise Perturbation
> >
> > We agree that our current discussion of the proposed multi-level noise–perturbed guidance is mostly empirical, and we will add a brief theoretical intuition in the revision.
> >
> > At a high level, our motivation comes from the way GAN generators are supervised, compared to other generative models. In diffusion or autoregressive models, the *input* and *output* at each step are tightly coupled by the objective (e.g., step-wise denoising or next-token prediction), so the model is explicitly forced to maintain a strong, layer-wise relationship between the noise input and the final sample. By contrast, a standard GAN generator typically receives a nearly “constant-style” latent input, and the loss only constrains the **final image** through the discriminator. As long as the last few blocks can produce samples that fool the current discriminator, there is little pressure for earlier blocks to carry informative structure; optimization can effectively push most of the representational burden into the blocks closest to the output.
> >
> > The multi-level noise–perturbed guidance is designed to partially compensate for this asymmetry. By introducing and supervising **noise signals at multiple levels** (with different noise strengths as detailed in the method section), we encourage the generator to respond meaningfully to these signals throughout its depth, rather than relying predominantly on the final blocks. In other words, the generator is nudged toward a more “coarse-to-fine” usage of its layers, where different depths are exposed to different noise scales and cannot simply collapse to near-constant mappings. We will clarify this intuition and its connection to the design of the guidance term in the revised manuscript.
> >
> > Moreover, we include this disccusion and perform additional experiments about the effects of MNG in Appendix A.7.

---

### Official Review · Reviewer_bP1g · 2025-11-03

**Soundness:** 3
**Presentation:** 3
**Contribution:** 2
**Rating:** 4
**Confidence:** 3

**Summary:**

This paper is the first to conduct a systematic study on the scalability of Generative Adversarial Networks (GANs) and proposes a framework named GAT. This framework combines two key design choices: training in the compact latent space of a VAE and using a pure Transformer architecture for both the generator and discriminator. To address issues such as underutilization of early layers and optimization instability that arise when scaling up the model, the authors propose two simple and effective solutions: a mechanism that provides lightweight supervision to the intermediate layers of the generator through multi-level noise perturbation, and a rule for adaptively adjusting the learning rate based on model width. Experiments show that GAT can be trained stably across a wide range of model capacities (from S to XL). Its largest model, GAT-XL/2, achieves state-of-the-art performance (FID 2.96) in class-conditional generation on ImageNet-256 after only 40 training epochs, demonstrating a 6x improvement in training efficiency compared to previous strong baselines.

**Strengths:**

+ This is the first work to systematically scale GANs across various Transformer model capacities and demonstrate stable performance improvement with increasing scale, laying the foundation for research into GAN scalability.

+ It effectively addresses the core challenges of scaling GANs through straightforward techniques (such as intermediate layer supervision and adaptive learning rates). The methods are simple and easy to implement, offering high practical value.

+ It achieves highly competitive performance in single-step generation tasks while retaining the inherent advantages of GANs, such as fast inference and latent space manipulability, proving the effectiveness of the framework.

**Weaknesses:**

- Despite training in latent space, the computational cost for the largest model remains very high, requiring 8 A6000 GPUs for 12 days, which limits its reproducibility and accessibility for further exploration.

- The experimental scope is relatively limited, focusing only on class-conditional generation on the ImageNet-256 dataset. It lacks validation on higher resolutions, cross-dataset generalization, or other conditional modalities (such as text-to-image generation).

- Although claimed to be a "pure Transformer" architecture, the generator incorporates style modulation techniques inspired by StyleGAN, which somewhat deviates from the original intention of "preserving the plain Transformer design."

**Questions:**

- Why does scaling the discriminator lead to greater performance improvements than scaling the generator? Does this imply that the key to enhancing GAN performance in the future lies in strengthening the discriminator?

- Can the adaptive learning rate rule proposed in the paper be further extended to form more comprehensive scaling laws for GANs, such as simultaneously considering other dimensions like model depth and batch size?

- Beyond class-conditional image generation, how can the GAT framework be applied to broader generative tasks, such as text-to-image generation? What new scalability challenges might arise in these tasks?

---

> ### Author Response · Authors · 2025-11-28
>
> Thanks for your valuable comments.
>
> ---
>
> ### 1. Experiments on a broader generative task (Text-to-image generation)
>
> In response to the reviewer’s comment, we added a text-to-image experiment on MS-COCO at 256 resolution.
>
> To this end, we build MM-GAT as follows. MM-GAT conditions GAT on CLIP text features using a simple scheme where the CLIP word tokens are fed into the generator (concatenated with the input positional encoding) and the [eot] embedding is used as a conditional vector, following an MM-DiT [bP1g-1]–style setup. Even with this simple conditioning mechanism and only minimal tuning under the short rebuttal timeline, the resulting MM-GAT already achieves competitive FID and outperforms prior 1-NFE GAN baselines.
>
> This indicates that GAT naturally extends to text-conditioned generation, and we expect that more careful tuning and refined conditioning designs could further improve performance.
>
>
> - **Text-to-image generation on MS-COCO at 256 resolution. We report FID and the number of function evaluations (NFE) at sampling time (lower is better). Ours achieves comparable FID with a lower NFE compared to diffusion models.**
> | Method           | Type               | NFE | FID   |
> |------------------|--------------------|-----|-------|
> | Diffusion models |                    |     |       |
> | VQ-Diffusion     | Discrete diffusion | 100 | 19.75 |
> | Frido            | Diffusion          | 200 | 8.97  |
> | U-Net            | Diffusion          | 50  | 7.32  |
> | U-ViT-S/2        | Diffusion          | 50  | 5.95  |
> | U-ViT-S/2 (Deep) | Diffusion          | 50  | 5.40  |
> | GAN models       |                    |     |       |
> | AttnGAN          | GAN                | 1   | 35.49 |
> | DM-GAN           | GAN                | 1   | 32.64 |
> | DF-GAN           | GAN                | 1   | 19.32 |
> | XMC-GAN          | GAN                | 1   | 9.33  |
> | LAFITE*          | GAN                | 1   | 8.12  |
> | MM-GAT (Ours)    | GAN                | 1   | 7.98  |
>
>
>
> Details and additional samples are provided in Appendix A.9.
>
>
>
> [bP1g-1] Esser, Patrick, et al. "Scaling rectified flow transformers for high-resolution image synthesis." Forty-first international conference on machine learning. 2024.
>
> ---
>
> ### 2. Concern about the high computational cost
>
> We agree that the largest GAT configuration is computationally demanding, mainly because, unlike a purely generative diffusion model, it jointly updates both a transformer generator and a strong discriminator at every iteration. However, high per-iteration cost does not directly translate to poor time-to-quality. For example, DiT-XL/2 on ImageNet-256 requires about 1400 epochs (36 days on 8×A100 GPUs) to reach FID 2.27, whereas our GAT-XL/2 attains ***a better FID-50K of 2.18 after only 60 epochs*** (trained on 8×RTX A6000 GPUs). Thus, for comparable or better image quality, GAT actually needs fewer total passes over the data and substantially less wall-clock time than a diffusion transformer.
>
> We also note that most of this cost is dominated by the discriminator, which repeatedly evaluates real/fake samples and the gradient-penalty term. As discussed in the paper, we expect that using a pretrained feature extractor as the discriminator backbone (in the spirit of projected GANs) and fine-tuning only the latter layers could significantly reduce this overhead, and systematically exploring such pretrained discriminators for GAT is an important direction for future work.
>
> We include this discussion in A.6 in Appendix.
>
> ---

---

> > ### Author Response · Authors · 2025-11-28
> >
> > ### 3. Concern about the pure transformer architecture
> >
> > Our use of the term *“pure Transformer”* is intended to describe the **backbone architecture**, not the complete absence of any conditioning or modulation. Concretely, our generator follows a standard ViT/DiT-style design: patchified inputs and a simple stack of self-attention + FFN blocks with normalization and residual connections, without any convolutional or hybrid CNN components.
> >
> > The only StyleGAN-inspired element is a **lightweight style modulation**, implemented as a per-channel affine transform on normalized features. This is conceptually similar to conditional LayerNorm / FiLM-style modulation commonly used in Transformer-based generative models, including DiT-type architectures [bP1g-2] that are still referred to as “pure Transformer” backbones despite employing conditioning mechanisms. We will revise the text to make this explicit, and describe our model more precisely as a *pure Transformer backbone with StyleGAN-inspired modulation*.
> >
> >
> > [bP1g-2] Peebles, William, and Saining Xie. "Scalable diffusion models with transformers." Proceedings of the IEEE/CVF international conference on computer vision. 2023.
> >
> > ---
> >
> > ### 4. Why does scaling the discriminator lead to greater performance improvements than scaling the generator?
> >
> > We attribute this mainly to the **asymmetric roles of G and D** in GAN training.
> >
> > The generator never sees real data directly; it only learns through the **training signal provided by the discriminator**. Thus, the overall performance is effectively *bottlenecked* by how well D can model the data distribution and provide informative gradients.
> >
> > When we scale up D, it learns a **sharper, more expressive real–fake decision boundary**, which yields higher-quality gradients for G (better coverage of modes, finer details, fewer noisy signals). Even a large G cannot fully exploit its capacity if the critic is weak, whereas a stronger D immediately improves the supervision signal that drives all updates to G. This is why, in our experiments, scaling the discriminator leads to more pronounced gains than scaling the generator alone.
> >
> > We will clarify this in the revised manuscript.
> >
> > ---
> >
> > ### 5. Extending the Adaptive Learning-Rate Rule to Comprehensive GAN Scaling Laws (Depth and Batch Size)
> >
> > We believe that our rule can indeed be extended into a more comprehensive scaling law that also accounts for depth and batch size. The underlying argument (keeping the per-step change in the network output approximately constant) is not specific to width. For example, in deep transformer-style residual networks, stacking $L$ blocks effectively increases the cumulative step size, so one could combine our width-based rule with an additional $\sqrt{L}$-type factor (or equivalently, depth-dependent Layerscale initialization) to preserve a similar overall update magnitude. Likewise, along the batch axis, our rule can be composed with the standard linear or square-root learning-rate scaling heuristics commonly used for large-batch transformer training.
> >
> > However, in this work we intentionally focused on width for clarity and empirical support. We performed preliminary experiments where we adjusted Layerscale initialization to compensate for depth, but within our depth range this did not lead to a noticeable benefit in stability and performance, so we did not further pursue depth-dependent tuning. In addition, all of our main experiments use a fixed batch size (512), so we do not yet have systematic evidence to justify baking batch-size dependence into the rule. We therefore view extending the rule jointly over width, depth, and batch size as a promising direction for future work.
> >
> > We include this discussion in A.12 in Appendix.
> >
> > ---

---

### Official Review · Reviewer_v5dc · 2025-11-03

**Soundness:** 4
**Presentation:** 4
**Contribution:** 3
**Rating:** 6
**Confidence:** 5

**Summary:**

The paper presents a scalable GAN framework that includes architectural changes, loss function choices, learning in the latent space, and scaling laws that enable stable GAN training. Unlike other generative models like Diffusion models, GANs enable single step generation and semantic latent spaces due to a compact noise dimension. The authors present positive results with these modifications and acheive SoTA in FID for ImageNet 256x256.

**Strengths:**

1. The paper addresses fundamental challenges in scaling up GAN training that actually works and achieves SoTA FID scores on ImageNet 256x256. This is a significant contribution and strength of the paper.
2. The authors' present thorough ablations and the narrative of the paper builds from the challenges to the final SoTA results while focusing on scalability of various components, including, architecture, loss functions, auxiliary losses, etc.
3. The presented model GAT-XL/2 achieves a SoTA FID score on class-conditional generation on ImageNet 256x256
4. The motivation for reviving GANs over SoTA Diffusion Models is made quite clear in Section 2.1, helping readability.

**Weaknesses:**

Minor:
1. Please use abbreviations only after they have been expanded at least once before. E.g. "GAT" in the Abstract.
2. On Line 137, the quote `Layerscale` -- can you please provide citations or a definition of what is meant by Layerscale? Is it referring to the scale factor in the BN layer - if so, calling it the `scaling factor in BN layer` would be much more clarifying.
3. It is unclear to the reader how the VAE fits in to this scalable GAN framework since the "VAE Latent" word is used from the very beginning of the paper. Only later does it clear up that a pretrained VAE model is being used to apply GAN in its latent feature space. Other than that, there is not any connection conceptually to VAE/Variational Bounds, etc. Please clarify this early on to improve readability. Thank you.
4. Please include SD-VAE's performance in Table 1 too so it is clear how much of the gains are due to the latent VAE representation.
5. Line 376, please cite LPIPS.
6. Under Related Works, for GANs various works like SNGAN, WGAN, WGAN-GP have been cited, that are similar. I suggest citing a family of models that use Discontinuous Functions in their Discriminator (GraN-GAN and Gradient Normalization) by Wu et al. and Bhaskara et al. (See Papers [1] and [2] below).


Major:
1. On Line 86, the authors claim that the larger the network, the rapid are the changes in the outputs. Please clarify this statement and back it up with citations or empirical / toy experiment with plots. What are the "outputs" here? Do they refer to the discriminator's logits or the generators image tokens? How to measure "speed" of change of the outputs?
2.  One Line 152, the authors say, "the early layers of the generator remain largely inactive." The reader is instantly led to question where the evidence for this is. So please cite your ablation study in Fig 4a even though it is still early in the paper -- than leading the reader to wrongly believe early on in the paper that it does not present enough evidence to claims.
3. Fig 4a does not seem sufficient to argue that early layers are not active in the generator to motivate for the layer-wise aux losses. Can a quantitative plot be presented. Example: Average abs rate of change of weights per layer over the course of training and showing that the early layers don't change much unless the MNG aux losses are used. Something quantitative would be really helpful. For instance, along the lines of Figure 3 in Bhaskara et al. [1].
4. Equation 4 is a hueristic for scaling the learning rate. Please back this up with an empirical quantitative plot. In the ablation (Sec 3.3) only qualitative arguments are given. Please show numbers to back up the claim for this scaling law. Also clarify if this scaling law applies to both Generator and Discriminator or only one of them? Also please include the initial LRs used for D and G for your SoTA model.


## References:
===
[1] Bhaskara et. al., GraN-GAN: Piecewise Gradient Normalization for Generative Adversarial Networks (WACV 2022) - https://openaccess.thecvf.com/content/WACV2022/html/Bhaskara_GraN-GAN_Piecewise_Gradient_Normalization_for_Generative_Adversarial_Networks_WACV_2022_paper.html.

[2] Wu et al., Gradient Normalization for Generative Adversarial Networks (ICCV 2021) - https://ieeexplore.ieee.org/document/9710427.

**Questions:**

In addition to the weaknesses above, I have the following additional questions:

1. Why was the pretrained VAE model used to extract the latent features for learning? Why not VGG/ResNet hidden features or BigGAN's hidden latent features? This is not clear and a clarification would greatly improve the cohesiveness of the paper
2. What is the module that remaps the generated latents back into the image space? This was not clear. Please describe this decoder mapping architecture.
3. In Figure 2, the list [x,x,x,x,] is labeled as copies of real images. This is confusing. Did you mean copies of the VAE latents of the real images instead? Please clarify.
4. It is not clear why the authors do not have any experiments on Unconditional Generation. Why is this? Is this because it is more challenging and the authors could not make it work (please share any negative results, as this strengthens the paper).

I am inclined to update my score given the authors can clarify and address some of the points raised. Thank you.

---

> ### Author Response · Authors · 2025-11-28
>
> Thanks for your valuable comments.
>
> ---
>
> ### 1. Clarification of the changes in the outputs
>
> We will clarify Line 86 as follows:
>
> - **Outputs** refer to any forward pass of the network $f_\theta(x)$ or input $x$, including discriminator logits and generator outputs (synthesized latent tokens).
> - **“Speed of change”** denotes how much $f_θ(x)$ changes after **one** optimization step, e.g. $E_x[∥f_{θ_{t+1}}(x)−f_{θ_t}(x)∥]$
>
>
> We also add a short mathematical argument showing that, under a fixed learning rate, wider networks tend to induce larger changes in these outputs per step, which motivates our width-aware learning-rate rule.
>
> #### Why wider networks change outputs faster (simple argument)
>
> It is standard in deep learning to assume that intermediate activations are roughly standardized on each channel, an assumption further supported by normalization mechanisms like LayerNorm and RMSNorm.
> So we can model hidden vectors as
> $x \in \mathbb{R}^C,\quad x_i \sim \mathcal{N}(0,1).$
>
> Consider a single linear layer with scalar output:
> $f_\theta(x) = w^\top x.$
>
> If the loss gradient on the output is $(g)$ and the learning rate is $(\eta
> )$, one SGD step gives
> $w' = w - \eta g x.$
>
> Evaluating at the **same** input \(x\),
> $f_{\theta'}(x) = w'^\top x = f_\theta(x) - \eta g \|x\|^2,$
>
> so the per-step change is
> $\Delta f(x) \coloneqq f_{\theta'}(x) - f_\theta(x) = -\eta g \|x\|^2.$
>
> For channels with unit variance,
> $\|x\|^2 = \sum_{i=1}^C x_i^2 \approx C,\quad
> \mathbb{E}[\|x\|^4] \approx C^2,$
>
> and hence
> $\sqrt{\mathbb{E}[\Delta f(x)^2]} \approx |\eta g| C.$
>
> In words: **with a fixed learning rate**, the typical per-step change in the output grows roughly linearly with the width \(C\). Since the same mechanism applies to every layer in the network, the per-step change of **all intermediate layer outputs** increases with $(C)$, and the resulting end-to-end change in $(f_\theta(x))$ inherits the same $(\mathcal{O}(C))$ scaling. This is what we meant by “larger networks change their outputs more rapidly.” To keep this functional update scale roughly constant across widths, we therefore use
> $\eta \propto \frac{1}{C}.$
>
> We added this argument in A.11 in Appendix.
>
> ---
>
> ### 2. Request for quantitative evidence of early-layer inactivity
>
> We thank the reviewer for raising this point. In the main paper (Fig.~4b), we *quantitatively* analyze early-layer inactivity by ablating each transformer block, regenerating images, and measuring perceptual distance. Without MNG, removing early blocks causes only minor changes in the outputs, even though these eariler layers should carry global structure, indicating that they are underutilized.
>
> To further support this, we generate 64 images per class and, for each block, measure (1) cosine similarity between its input and output features and (2) cosine similarity across different input noises $z$, averaged over tokens and images. For the model **without** MNG, both similarities are very close to 1 in early layers, showing that they barely modify features and are weakly influenced by $z$. With MNG, these similarities are substantially lower in early blocks, confirming that MNG increases early-layer activity and makes them more responsive to the input noise.
> For visualization, we report the according plot in Fig.9 in Appendix.
>
> Following the reviewer’s suggestion, we also inspected per-layer gradient magnitudes over training, but did not observe a clear systematic difference. Since gradient norms are only an indirect proxy for contribution to generation after training, we emphasize activation-space measurements instead. We include this discussion in Appendix A.7.
>
> ---
>
> ### 3. Quantitative ablation of Eqn. 4 and implementation details
>
> We apologize for the inconvenience.
> We omit the referencing the according plot, which is Fig.5b.
> The exact FID-5K at 50K iteration is reported in Fig.5b (y-axis), and we added exact number in Table.4 (Appendix A.3).
> For our learning rate scaling rule, we apply it to both the generator and discriminator. Finally, we use 1.33e-4 for both networks as noted in Appendix A.1, as we set the base learning rate for GAT-S/2 as 4e-4.
>
> Moreover, we added the further disccusion about Eqn.4 in Appendix A.11 and comment 1.
>
> ---

---

> > ### Author Response · Authors · 2025-11-28
> >
> > ### 4. Confusion about pretrained VAE and its latent and decoder
> >
> > Thank you for pointing out the potential confusion about the term, VAE.
> > In our framework, *“VAE latents”* are the bottlneck feature of a **frozen pretrained image autoencoder** used purely as an image **tokenizer/de-tokenizer**: it compresses a 256×256 RGB image into a 32×32 spatial latent grid and reconstructs it back, but cannot synthesize images from noise.
> > These choices follow conventions that are now widely adopted in modern large-scale generative models, including diffusion models such as Stable Diffusion [v5dc-1].
> > The encoded latents are designed to preserve as much image information as possible under compression, so real and generated samples share a common, approximately invertible representation. We train the GAN **entirely in this compressed latent space** for efficiency (far fewer tokens and FLOPs), which makes it feasible to scale generator/discriminator width and depth.
> >
> > This is conceptually different from conventional feature embeddings: VGG/ResNet features are high-level, task-oriented, and non-invertible, while BigGAN’s hidden features are internal activations without an encoder, so real images cannot be mapped consistently and capacity would be bottlenecked by BigGAN itself. In the revised introduction, we will explicitly state that “VAE latent” always refers to this frozen tokenizer representation and that the VAE’s sole role is to map images to and from a compact latent grid.
> >
> >
> > [v5dc-1] Rombach, Robin, et al. "High-resolution image synthesis with latent diffusion models." Proceedings of the IEEE/CVF conference on computer vision and pattern recognition. 2022.
> >
> > ---
> >
> > ### 5. No experiments about the unconditional generation
> >
> > Our main experiments on ImageNet were designed to study how a transformer-based GAN scales in a complex, diverse, large-scale setting. To additionally verify the feasibility of unconditional training, we also train GAT-B/2 on FFHQ-256 for 25M images (i.e., 50K iterations with a batch size of 512). We emphasize that a pure transformer operating in the latent space of a fixed tokenizer generally performs poorly on small datasets, making this configuration particularly challenging. Moreover, FFHQ is a relatively small and highly curated dataset of aligned human faces, while our generator is an almost “unbiased” transformer with weak built-in inductive biases; this mismatch between a strongly biased data distribution and a flexible architecture can naturally incur a slight loss in sample efficiency compared to more specialized designs. Nevertheless, when compared against a DiT-B/2 baseline trained under the same protocol, our model achieves an FID of ***9.74*** versus ***10.49*** for DiT-B/2, indicating that GAT can successfully support unconditional generation and attains competitive performance with diffusion transformers.
> >
> > We include this experiment and qualitative examples in A.10 and Fig.12 in Appendix.
> >
> > ---
> >
> > ### 6. Minor comments and questions
> >
> > We carefully revise the manuscript, including missing references such as LPIPS and other GAN-relevant literature in belows.
> >
> > - Layerscale denotes the scaling of the output of residual blocks, which is generally initialized by small values to stabilize the training of deep transformers [v5dc-2].
> > - For copies of real images in Fig. 2, we aim to refer to the VAE latent tokens before applying the noise perturbation. We will revise it by explicitly noting them as a VAE latent in the Figure.
> >
> > [v5dc-2] Touvron, Hugo, et al. "Going deeper with image transformers." Proceedings of the IEEE/CVF international conference on computer vision. 2021.
> >
> >
> > ---

---

### Author Response · Authors · 2025-11-28
**Brief summary of rebuttal**

We thank the reviewers and the AC for their thoughtful and constructive feedback. Our paper presents a systematic study of how to scale transformer-based GANs in a VAE latent space. In the revision, we substantially broadened the experimental scope (higher resolution, new datasets and modalities, unconditional setting), added new analyses for early-layer inactivity and the width-aware learning-rate rule, and clarified the training cost, time-to-quality, and inference efficiency. Overall, we believe the main concerns have been addressed and that the paper now provides a solid and complete study of scalable transformer-based GANs in the VAE latent space.

---

### Strengths highlighted by reviewers

- **First systematic scaling study of transformer-based GANs**, demonstrating stable improvements from S to XL and reliable training of large GANs. (v5dc, bP1g, Y3NZ, dMUD)

- **Strong ImageNet-256 results**: GAT-XL/2 achieves state-of-the-art / highly competitive FID with single-step generation and far fewer epochs than diffusion/flow baselines. (v5dc, bP1g, dMUD)

- **Simple, practical techniques**: Multi-level Noise-perturbed Guidance (MNG), width-aware learning-rate scaling, and VFM alignment (REPA), which are easy to implement and likely useful beyond this paper. (bP1g, dMUD)

- **Pure transformer backbones** for both G and D, with a clear motivation for revisiting GANs (single-step inference, editable latent space). (dMUD)

---

### How we addressed the main concerns

1. **Experimental scope was too narrow (mostly ImageNet-256, class-conditional). (v5dc, bP1g, Y3NZ, dMUD)**

    We added:
    - Higher-resolution experiments on ImageNet-512,
    - A text-to-image experiment on MS-COCO-256 with a multimodal GAT variant, and
    - Unconditional generation on FFHQ-256.

    These results show that GAT scales favorably to higher resolutions, extends to multimodal text-conditional generation, and also transfers to unconditional image generation. (See items (2)–(3) in the “Summary of our responses” below and Appendices A.8–A.10.)

2. **Request for detailed training cost / time-to-quality / NFE comparisons vs DiT-style diffusion. (bP1g, Y3NZ, dMUD)**

    We added detailed comparisons between GAT-XL/2 and DiT-XL/2 on ImageNet-256, including epochs, FID, wall-clock time, NFE, and memory. These results show that GAT achieves better FID with far fewer epochs and roughly **×200 faster sampling** due to one-step generation, while requiring **lower overall training compute for comparable quality**. (See items (1) and (6) and Appendices A.5–A.6.)

3. **Insufficient evidence and analysis for early-layer inactivity and the effect of Multi-level Noise-perturbed Guidance (MNG). (v5dc, Y3NZ, dMUD)**

    We added quantitative analyses of MNG, measuring per-block feature change (input–output similarity) and sensitivity to input noise, with and without MNG. The results confirm that, without MNG, early transformer blocks contribute little to the final image, whereas MNG activates early layers and leads to a clearer coarse-to-fine synthesis pattern, together with improved FID. (See item (5) and Appendix A.7.)

4. **Width-aware learning-rate rule not fully justified; requires further discussion of its relation to depth/batch scaling. (v5dc, bP1g, dMUD)**

    We elaborated the intuition and mathematical justification for our width-aware LR rule, clarifying the assumption of approximately standardized per-channel activations and showing why outputs change faster in wider networks. We also added discussion of how the rule relates to depth and batch-size scaling. (See item (7) and Appendix A.11.)

5. **Dependence on tokenizer choice and on VFM alignment (REPA), and the need for more evidence on latent-space quality (e.g., applicability to editing). (dMUD)**

    We show that GAT benefits from stronger tokenizers but does not rely on a particular one: with the FLUX-e2e tokenizer, FID improves substantially over SD-VAE (from **4.60** with SD-VAE to **3.73** with FLUX-e2e). We also analyze the VFM alignment objective, demonstrating that it adds less than **10%** overhead per iteration and consistently improves FID across model scales. Finally, we add qualitative latent-space editing results (GANSpace and StyleCLIP) showing that the learned latent space remains smooth and semantically meaningful. (See items (8)–(9) and (11), and Appendices A.13–A.15, A.18.)

---

Overall, after these revisions we believe that:

- the paper now provides a **comprehensive and well-validated study of scaling transformer-based GANs** in VAE latent space,
- **all major concerns have been substantively addressed**, regarding experimental scope, analysis of MNG and the LR rule, and training cost, and
- the work offers a strong, practically useful baseline for scalable one-step generation.

We therefore hope that the AC will consider the revised version closer to the accept side of the decision spectrum.

---

> ### Author Response · Authors · 2025-12-01
> **Summary of our responses**
>
> ---
>
> ### *Summary of our responses*
>
> 1. Extended ImageNet-256 training of GAT-XL/2, showing ***FID-50K 2.18 at 60 epochs***, significantly outperforming MeanFlow-XL/2 (one step, FID-50K of 3.43) and also surpassing DiT-XL/2 (250 steps, FID-50K of 2.27) which requires around ***1400 epochs*** on ImageNet-256 to reach comparable quality, while achieving approximately ***×200 faster inference*** (Appendix A.5–A.6).
>
> 2. (bP1g,Y3N,dMUD) New generative experiments on ***higher-resolution*** ImageNet-512, ***text-to-image*** MS-COCO (Appendix A.8-9).
>
>     - On ImageNet-512, GAT-XL/2 reaches ***FID-50K 4.04 at 15 epochs***, comparable to the 256×256 result at 20 epochs, indicating successful and efficient training on high-resolution data.
>
>     - On MS-COCO-256, a simple multimodal GAT variant achieves ***FID 7.98***, already competitive and suggesting strong potential for further improvements with more careful tuning.
>
> 3. (v5dc) Experiments on ***unconditional generation*** FFHQ-256 (Appendix A.10)
>
>     - GAT-B/2 achieves ***FID 9.74 vs. 10.49*** for DiT-B/2 at 25M images on FFHQ-256, demonstrating that GAT transfers well to unconditional generation.
>
> 4. (dMUD) ***Qualitative comparison*** with one-step generative models (StyleGAN-XL, MeanFlow) (Appendix A.14)
>
>     - Visual comparisons show that GAT-XL produces higher-quality samples than prior one-step GANs and flow/diffusion models.
>
>
> 5. (v5dc,Y3NZ,dMUD) ***Additional analysis of Multi-level Noise-perturbed Guidance (MNG)***, clarifying its motivation and training dynamics (Appendix A.7).
>
>     - Added intuitive explanation of MNG and further experiments measuring the effect of each generator block (feature similarity between block input/output and sensitivity to input noise), showing that MNG activates early layers more effectively.
>
> 6. (Y3NZ,dMUD) Detailed ***training-time and NFE comparisons*** between GAT and DiT-style diffusion transformers on ImageNet-256, including wall-clock breakdowns and compute-efficiency table (Appendix A.6).
>
>     - Analysis shows that, for a generator of comparable size and FID, GAT requires lower overall training compute than DiT and achieves roughly ***×200 faster inference*** at sampling time.
>
> 7. (v5dc,bP1g,dMUD) ***Deeper study of the width-aware learning-rate rule***, including intuition and potential extensions to depth and batch size (Appendix A.11).
>
>     - We elaborate ***the intuition and mathematical justification*** for our width-aware LR rule, and discuss how LR scheduling can be combined with depth- and batch-size–dependent scaling.
>
> 8. (dMUD) Robustness of learned latent space via ***image editing*** (unsupervised and text-driven editing) (Appendix A.13).
>
>     - Demonstrated that GAT-XL/2 latents on ImageNet-256 support meaningful edits using GANSpace (unsupervised directions) and StyleCLIP (text-driven editing).
>
> 9. (dMUD) ***Effects of the choice of tokenizers*** by training GAT on different tokenizer (Appendix A.15).
>
>     - Using the state-of-the-art FLUX-e2e tokenizer, GAT-L/2 improves from ***FID 4.60 (SD-VAE)*** to ***FID 3.73 (FLUX-e2e)*** at 20 epochs on ImageNet-256, showing stable training and better performance with a stronger tokenizer.
>
> 10. (dMUD) ***Formal scaling law*** with ***seed-robustness*** results (Appendix A.16–A.17).
>
>     - We provide a log–log scaling law relating training compute to FID, and show seed-robustness by reporting similar FID values across multiple random seeds.
>
> 11. (dMUD) ***Detailed analysis of the VFM alignment objective*** including its computational cost, effectiveness across model size, and its effects when fine-tuning (Appendix A.18).
>
>     - We show that the VFM alignment loss adds less than 10% overhead per iteration, consistently improves FID for GAT-S/2 and GAT-B/2, and its behavior when resuming training without VFM alignment objective.
>
>
> ---

---

### Meta-Review · Area_Chair_584W · 2026-01-06

**Summary:**

- Experiments are mostly restricted to class-conditional ImageNet-256; no evaluation on higher resolutions, other datasets, unconditional generation, or different modalities (e.g., text-to-image). This raises questions about generality and scalability claims.

- Core design choices (e.g., Multi-level Noise-perturbed Guidance (MNG), adaptive learning rate scaling) lack strong theoretical justification, ablation, or quantitative evidence. Some heuristics are unclear or insufficiently motivated.

- Despite latent-space training, the largest model requires extremely high compute (12 days on 8×A6000 GPUs), limiting reproducibility and broader adoption. Transformer-based architectures exacerbate memory issues.

**Reviewer Concerns:**

- The authors added text-to-image experiments on MS-COCO (256×256) using MM-GAT conditioned on CLIP embeddings, showing competitive FID scores and lower function evaluations (NFE) than prior GAN baselines. They also trained GAT-XL/2 on ImageNet-512, demonstrating comparable FID to 256×256 with fewer epochs, suggesting the method can handle higher resolutions. The rebuttal emphasizes that the main paper focused on scaling model capacity rather than resolution, which clarifies the experimental design rationale. Thus, the first issue is largely addressed.

- The rebuttal improves conceptual explanation and points to additional details in appendices, but it does not provide the clear, quantitative, causal evidence or theoretical grounding for Eqn. 4 and MNG. Essentially, the rebuttal mostly restates motivation and references existing plots.

- The rebuttal shows that GAT is more efficient in total time-to-quality and inference than diffusion transformers, but it does not fully alleviate concerns about high computational requirements and limited accessibility, because the largest configuration is still very expensive to train, and the proposed mitigations remain speculative for future work.

**Reviewer Scores:**

All reviewers are likely to keep their current ratings.

---

### Decision · Program_Chairs · 2026-01-26

Reject